# VideoREPA: Learning Physics for Video Generation through Relational Alignment with Foundation Models

**Xiangdong Zhang**[1‡]**, Jiaqi Liao, Shaofeng Zhang**[1]**,**
**Fanqing Meng**[1]**, Xiangpeng Wan**[2]**, Junchi Yan**[1]**, Yu Cheng**[3†]
[1]School of AI & School of CS, Shanghai Jiao Tong University
[2]NetMind.AI, [3]The Chinese University of Hong Kong

## Abstract

Recent advancements in text-to-video (T2V) diffusion models have enabled high-fidelity and realistic video synthesis. However, current T2V models often struggle to generate physically plausible content due to their limited inherent ability to accurately understand physics. We found that while the representations within T2V models possess some capacity for physics understanding, they lag significantly behind those from recent video self-supervised learning methods. To this end, we propose a novel framework called VideoREPA, which distills physics understanding capability from video understanding foundation models into T2V models by aligning token-level relations. This closes the physics understanding gap and enables more physics-plausible generation. Specifically, we introduce the Token Relation Distillation (TRD) loss, leveraging spatio-temporal alignment to provide soft guidance suitable for finetuning powerful pre-trained T2V models—a critical departure from prior representation alignment (REPA) methods. To our knowledge, VideoREPA is the first REPA method designed for finetuning T2V models and specifically for injecting physical knowledge. Empirical evaluations show that VideoREPA substantially enhances the physics commonsense of baseline method, CogVideoX, achieving significant improvement on relevant benchmarks and demonstrating a strong capacity for generating videos consistent with intuitive physics. Code and more video results are available at Project Page.

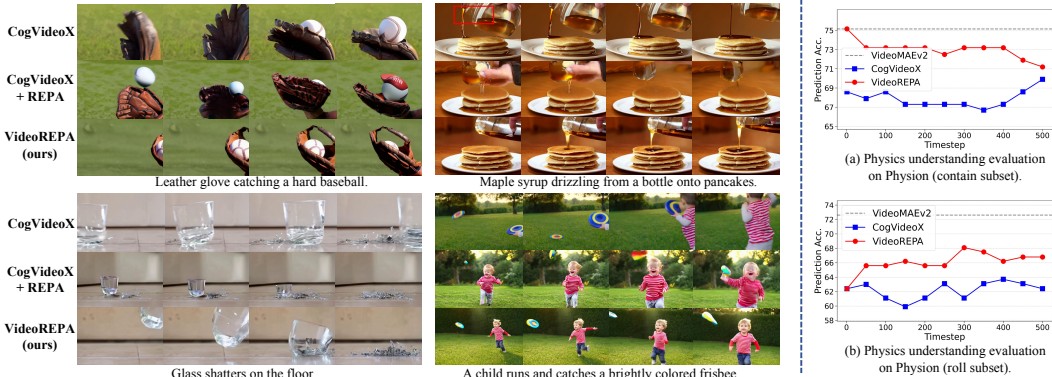

Figure 1: *Left:* Visual comparison of video generation results from CogVideoX [56] (baseline), CogVideoX finetuned with REPA [58], and our proposed VideoREPA. Red rectangles denote phenomena that violate physical commonsense for easier distinguish. **Our VideoREPA generates videos that most closely adhere to real-world physical laws.** *Right:* Evaluation of physics understanding on the Object Contact Prediction (OCP) task within the Physion benchmark [6]. The plots illustrate a significant gap in physics understanding between the SSL video encoder VideoMAEv2 and the T2V model CogVideoX. **The proposed VideoREPA substantially narrows this understanding gap.**

[‡]Interns at Shanghai AI Laboratory. [†]Corresponding author. `chengyu@cse.cuhk.edu.hk`.

39th Conference on Neural Information Processing Systems (NeurIPS 2025).

# 1 Introduction

Video diffusion models (VDMs) have recently gained significant attention, demonstrating remarkable advancements [1, 45, 24, 56, 31, 41]. These generative models are increasingly applied in diverse domains, including movie-level video generation [51], animation [49], and advertising [12]. However, a critical challenge remains: the physical plausibility (e.g., shape regularity and motion rationality) of videos generated by even state-of-the-art VDMs is often severely limited [4, 3]. While existing VDMs (e.g., VideoCrafter2 [9], CogVideoX [56], HunyuanVideo [24], Cosmos [1], Wan [45]) have shown improvements in physics capabilities, typically achieved by scaling training data, refining model architectures, or collecting higher-quality video-text pairs [4], these strategies have inherent drawbacks. The substantial expense of scaling datasets and the limited focus of current architectural designs on explicit physics modeling make significant advancements in physically plausible video generation through these avenues difficult and costly.

Current methods for enhancing the physical plausibility of generated videos can be divided into two main categories: simulation-based approaches [29, 27, 53, 52, 28, 60] and non-simulation-based approaches [46, 10, 55]. Simulation-based methods, which have seen significant development, typically integrate external physics simulators for guidance or direct generation. However, their effectiveness is constrained by the complexity of simulations and the challenge of modeling diverse open-domain phenomena, limiting their potential for creating powerful, general-purpose generative models. Non-simulation-based methods, on the other hand, have received less attention. For example, WISA [46] decomposes textual descriptions into physical phenomena and employs Mixture-of-Physical-Experts Attention. Yet, WISA only show improvement when training with dedicated collected dataset WISA-32K [46] where each relates to individual explicit physical phenomena. It struggles to generalize to open-domain data which is much easily to collected and scale. To this end, **this paper explores enhancing the physics-plausible video generation of T2V models using non-simulation strategies on open-domain datasets, aiming to achieve robust generalization and broad applicability.**

Regarding generating physical coherence videos, it is acknowledged that in generative modeling, improved understanding often benefits generation quality [22, 13, 54, 16, 23, 26]. DynamiCrafter [54], for example, exemplifies this by using enhanced visual encoding to improve its outputs. However, our evaluations on physics understanding benchmark Physion [6] (illustrated in Figure 1) reveal that the text-to-video diffusion model CogVideoX (2B) exhibits poor physics understanding ability, performing significantly worse than the much smaller self-supervised video understanding model, VideoMAEv2 (86M). This notable gap in physics comprehension motivates our core strategy: **to improve the physics understanding of VDMs by transferring inherent physics knowledge from ViFMs, and thereby enhance physically coherent video generation.**

Recent work has explored bridging the gap between foundation models and generative diffusion models, notably through Representation Alignment (REPA) [58, 25, 42], which enhances semantics in image diffusion models via feature alignment. Inspired by REPA, we investigate an approach to improve physical plausibility in video generation by aligning with ViFMs. However, directly applying REPA techniques to inject physics knowledge into text-to-video models **proves infeasible due to several critical distinctions**: First, the spatial focus of REPA is insufficient for crucial temporal dynamics in videos. Second, REPA targets from-scratch training acceleration, not knowledge transfer via finetuning pre-trained models. Third, its hard alignment mechanism can destabilize pre-trained VDMs during finetuning. Finally, VDM temporal latent compression adds further alignment complexity. Experiments in Figure 1 support this, showing that finetuning with REPA degrades the performance of CogVideoX significantly. See detailed discussions in Section 3.3.

To address these challenges and enhance physics in video generation by deepening physics understanding, we introduce **VideoREPA**. This method distills token-level relations, capturing dynamics from Video Foundation Models (ViFMs) and transferring them to VDMs, thereby improving the physical realism of generated videos. Our method performs well on open-domain video dataset OpenVid [35] without relying on the physics explicit dataset WISA-32K [46] which is much harder to collect. Specifically, we propose a Token Relation Distillation (TRD) loss that distills intra-frame spatial relations and inter-frame temporal dynamics from ViFM representations into VDMs via relational alignment, which closes the physics understanding gap as shown in Figure 1. Unlike standard REPA, our TRD loss employs a more moderate alignment mechanism tailored to overcome difficulties associated with fine-tuning. We argue that the physical plausibility of a video depends

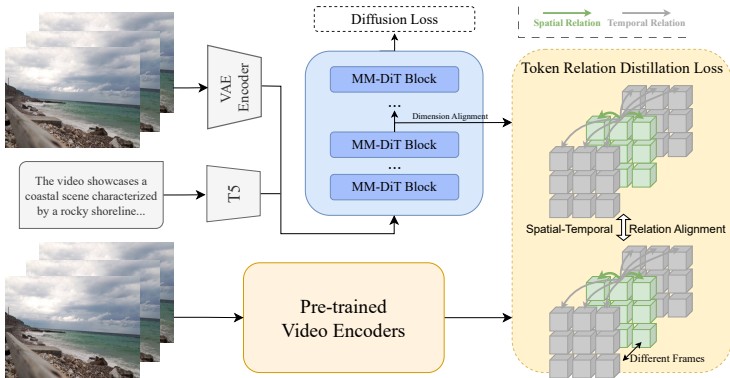

Figure 2: Overview of VideoREPA. Our VideoREPA enhances physics in T2V models by distilling physics knowledge from pre-trained SSL video encoders. We apply Token Relation Distillation (TRD) loss to align pairwise token similarities between video SSL representations and intermediate features in diffusion transformer blocks. Within each representation, tokens form spatial relations with other tokens in the same latent frame and temporal relations with tokens in other latent frames.

not only on the regular shape of objects (spatial dimension) but critically on the motion of subjects (temporal dimension); thus, focus of TRD extends beyond merely spatial alignment to capture these crucial temporal aspects. Our main contributions can be summarized as follows:

**1) We identify an essential gap in physics understanding between self-supervised ViFMs and T2V models.** We then propose VideoREPA, the first method to bridge video understanding models and T2V models, which closes understanding gaps and achieves more physically plausible generation.

**2) We introduce VideoREPA, a novel feature alignment framework for video generation.** It utilizes Token Relation Distillation loss to effectively distill physics knowledge from ViFMs through token-relational alignment, enabling VDMs to generate videos that better adhere to physical laws.

**3) The proposed TRD loss overcomes key limitations (detailed in Section 3.3) of directly applying REPA** [58] to the video generation domain, particularly for finetuning pre-trained models and capturing essential temporal dynamics.

**4) Quantitative and qualitative experiments show the superiority of our VideoREPA** over baselines like CogVideoX and other methods like WISA. VideoREPA achieves a state-of-the-art Physical Commonsense (PC) score of 40.1 on VideoPhy (**24.1% improvement over its CogVideoX baseline**) and significant physics enhancements on the challenging VideoPhy2 benchmark. Visualizations also confirm VideoREPA generates videos more consistent with physical laws than CogVideoX.

## 2   Related works

**Self-supervised learning for physics understanding.** Understanding physical interactions (e.g., predicting object trajectories [17] or movements [29]) is vital for applications like robotics [11] and autonomous driving [38]. Self-supervised learning (SSL), as a powerful tool for a wide range of understanding tasks including classification, segmentation, and detection [37, 8, 19, 21], leverages pretext tasks to pre-train models on large amounts of unlabeled data, thereby enhancing understanding ability and **yielding the powerful pre-trained models often called foundation models.** Recent studies [15, 44] highlight the potent physics understanding capabilities of Video Foundation Models, such as VideoMAEv2 [47], and V-JEPA [5], demonstrated through strong performance on physics benchmarks like Physion [6]. Notably, these specialized video models can outperform even large Multimodal Language Models like GPT4-V [36] on physics reasoning tasks. Building on the principle that strong understanding facilitates better generation [54], we investigate how to leverage the physics knowledge captured by video SSL models to enhance the physical realism of T2V generation.

**Physics plausible video generation.** While initial T2V model improved visual fidelity, motion, and realism using scaled data and advanced architectures [56, 45, 9, 7, 31, 59], recent studies [34, 3, 4, 32] highlight a major problem: physical plausibility remains poor even in state-of-the-art (SOTA) models. This realization has driven emerging research towards physics-aware generation, with many new methods proposed [46, 10, 27, 28, 52, 55, 60, 29, 33]. Some techniques, exemplified by

PhysAnimator [52], PhysGen [29] and MotionCraft [33] rely on direct physics simulation. However, these simulation-dependent methods are inherently limited by the simulator's scope and accuracy, making them less suitable for complex real-world scenarios. PhyT2V [55] uses MLLMs to refine prompts through multiple rounds of generation and reasoning, maximizing the physics potential in models but not adding new inherent knowledge. WISA [46] decomposes textual descriptions into physical phenomena and uses Mixture-of-Experts for different physics categories. However, it faces challenges in clearly defining physical components from diverse text prompts and its effectiveness is limited to specialized datasets (i.e., WISA-32K [46]) containing explicit physics phenomena and fails to generalize to open-domain datasets like Koala-36M [48], making large-scale data application difficult. This paper proposes VideoREPA, a training-based method that improves the physical realism of generated videos by aligning representations with those learned by video SSL models. VideoREPA features compatibility with open-domain datasets, enhancing its potential for wide adoption.

## 3 Methods

This section first covers preliminary on Latent Diffusion Models and REPA (Section 3.1). Subsequently, Section 3.2 analyzes the interplay between model understanding and generation capabilities, thereby establishing the motivation for our work. Building upon this, Section 3.3 presents our core contribution: VideoREPA, a novel framework featuring the **T**oken **R**elation **D**istillation (**TRD**) loss, distilling physics from video foundation models [47] by aligning token-level relations across both spatial and temporal dimensions. VideoREPA, **for the first time**, applies feature alignment to finetune pre-trained VDMs, leveraging relational distillation to incorporate spatial-temporal dynamics.

### 3.1 Preliminaries

**Latent Diffusion Models.** Latent Diffusion Models (LDMs) [40] typically operate in the latent space of a pre-trained Variational Autoencoder (VAE). They generate data by learning to reverse a forward diffusion process that gradually adds noise. Our VideoREPA framework is designed for finetuning transformer-based video LDM, i.e., CogVideoX [56]. The training objective is the mean squared error (MSE) between the added noise $\epsilon$ and the noise predicted by the model $\epsilon_\theta$:

$$\mathcal{L}_{\text{diff}} = \mathbf{E}_{t,z_0,\epsilon} \left[ \|\epsilon - \epsilon_\theta(\alpha_t z_0 + \sigma_t \epsilon, t)\|^2 \right] \tag{1}$$

where $z_0$ denotes the initial latent input (obtained by encoding video frames, e.g., via a 3D VAE), $\epsilon \sim \mathcal{N}(0, \mathbf{I})$ is sampled noise, $t$ is the diffusion timestep, $\alpha_t$ and $\sigma_t$ are schedule-dependent coefficients (e.g., $\alpha_t = \sqrt{\bar{\alpha}_t}, \sigma_t = \sqrt{1 - \bar{\alpha}_t}$), and $\theta$ represents the parameters of the denoising transformer.

**Representation alignment for generative models.** Representation alignment (REPA) [58] is a straightforward regularization method, demonstrating that the convergence speed of image diffusion model training process can be significantly improved by aligning representations $\mathbf{y}_*$ from encoders $f$ of pre-trained vision foundation models (e.g., DINOv2 [37]) with the internal representation. REPA distills the $\mathbf{y}_*$ of a clean image $\mathbf{x}$ into the denoising transformer representation $\mathbf{h_t}$ of a noisy input $\mathbf{x_t}$. Specifically, the $\mathbf{y}_* = f(\mathbf{x}) \in \mathbb{R}^{N \times D}$ and $\mathbf{h_t} = f_\theta(\mathbf{x_t})$ which will then be input to a trainable MLP $h_\phi$ for dimension alignment. The alignment loss can be formulated as that maximizes the token-wise feature similarities:

$$\mathcal{L}_{\text{REPA}} = -\mathbf{E} \left[ \frac{1}{N} \sum_{n=1}^{N} \text{sim}(\mathbf{y}_*^{[\mathbf{n}]}, h_\phi(\mathbf{h_t^{[\mathbf{n}]}})) \right] \tag{2}$$

The $\text{sim}(\cdot, \cdot)$ denotes a similarity metric (e.g., cosine similarity). This method reduces the semantic gap between $h_t$ and $y_*$, accelerating the training speed of DiT. The final loss is: $\mathcal{L} = \mathcal{L}_{\text{diff}} + \lambda \mathcal{L}_{\text{REPA}}$. REPA has been reported to speed up DiT training by over $17.5\times$. Subsequent works, such as VA-VAE [57] which improved VAE features, achieved $21\times$ speedups, and REPA-E [25] which proposed end-to-end training, reached up to $45\times$ acceleration for DiT.

### 3.2 Understanding vs. generation

Understanding refers to the ability of models to interpret input data and extract meaningful information, whereas generation involves generating novel data. A well-established principle in generative modeling is that enhancing understanding often leads to improved generation quality [22, 13, 54, 16].

DynamiCrafter [54], for example, improves video generation by leveraging CLIP and query transformer to better understand image conditions. Similarly, SmartEdit [22] and MGIE [13] utilize language models to enhance instruction understanding for more accurate image editing.

This principle motivates our work addressing the poor physics plausibility observed in leading T2V generation models [3]. We wonder whether physics plausibility in generation can be improved through deepening the model's understanding to physics and start by comparing the physics understanding abilities of large VDMs against specialized Video Foundation Models (ViFMs). Physion [6], a benchmark for evaluating physics understanding is used (detailed at Appendix C). As shown in Figure 1, CogVideoX (2B) demonstrates significantly weaker physics understanding compared to smaller VideoMAEv2 (86M) [47]. **This disparity highlights an opportunity**: bridging the physics understanding gap by transferring knowledge from capable ViFMs to powerful VDMs. Therefore, we propose a method based on representation alignment. The following section details our VideoREPA framework, which close the understanding the gap through TRD loss, as shown in Figure 1.

### 3.3 Token Relation Distillation loss

Introducing physics knowledge understanding into VDMs is non-trivial. Unlike text understanding, which can often be enhanced by incorporating a more powerful text encoder [22] since the text prompt is provided directly as input, text/image-to-video generation lacks direct video input during inference. Consequently, directly leveraging a powerful physics understanding video encoder to guide the generation process is generally infeasible.

Recently, REPA [58] has emerged as a method to bridge the semantic gap between pre-trained foundation models and diffusion models. By aligning internal representations during from-scratch training of diffusion models, REPA primarily aims to enhance semantics and accelerate training. This suggests a potential avenue for transferring physics understanding from capable ViFMs into VDMs. However, existing REPA [25, 42, 58] approaches are insufficient for effectively achieving this goal, particularly when finetuning pre-trained VDMs.

**The gap of applying REPA for physics enhancement in video generation model.** Though REPA and related methods [25, 58, 57, 42] build bridge between foundation and generation models, they become expired when aiming at bridging ViFMs and VDMs: **1) Shift in Focus (Spatial vs. Spatio-Temporal):** Existing REPA techniques predominantly focus on aligning *spatial* features within static images. However, physical plausibility in videos relies critically on *temporal dynamics*, i.e., the rational evolution of motion and interactions over time, in addition to the correct appearance within single frame at spatial dimension. Standard spatial alignment is insufficient to capture or enforce these crucial temporal dynamics. **2) Mismatch in Application Context (From-Scratch Acceleration vs. Finetuning for Knowledge Transfer):** Existing REPA approaches have primarily been validated and utilized for **accelerating** the *training of models from scratch* [57, 58], optimizing convergence speed. Our goal, however, is different: injecting specific knowledge (physics) into *already pre-trained* VDMs via finetuning. This needs further discussion and validation. **3) Mechanism Mismatch (Hard Alignment vs. Finetuning Stability):** REPA employs a direct feature similarity loss (e.g., cosine similarity) to train a DiT from scratch. When applied during finetuning, this "hard" alignment objective attempts to force potentially incompatible feature spaces—the latent space of pretrained and the SSL-optimized feature space of ViFM—to match directly. As demonstrated in our experiments (see Section 4.5), finetuning CogVideoX with standard REPA leads to significant degradation in semantic quality and overall coherence. We attribute this failure to the direct alignment disrupting the well-established internal representations in pre-trained VDMs. **4) Added Complexity (Temporal Compression in Latents):** Unlike typical image latents, VDM latent spaces often employ significant *temporal compression* (e.g., 4x in CogVideoX [56]). This adds complexity to the design alignment process, as the temporal granularity between the VDM's latent representation and the ViFM's feature representation may differ, requiring careful handling during alignment design.

These limitations collectively indicate that a naive application of standard REPA is unsuitable for enhancing the physical plausibility of pre-trained VDMs via finetuning. A different strategy is required—one that specifically addresses temporal dynamics, ensures stability during finetuning, and effectively manages differences between the VDM's latent space and the ViFM's feature space.

To address these challenges, we propose **VideoREPA**, a framework leveraging a novel **T**oken **R**elation **D**istillation (**TRD**) loss to distill physics understanding models for better physics plausible

generation. Instead of enforcing direct feature similarity (hard alignment), which proved unsuitable for finetuning (Section 4.5), TRD aligns the relational structure (i.e., pairwise token similarities) between the internal representations in VDMs and those of a capable Video Foundation Model (e.g., VideoMAEv2). This relational alignment provides a softer guidance suitable for finetuning, explicitly incorporates spatio-temporal dynamics by considering relationships both within and across frames, and issues arising from direct feature space incompatibility. Unlike prior REPA work focused on spatial alignment for image generative model acceleration [58, 57], VideoREPA represents, to our knowledge, the **first representation alignment method developed for finetuning VDMs to inject specific knowledge (physics contained in spatial-temporal dynamics)**, pushing beyond mere acceleration.

As shown in Figure 2, the TRD loss aligns pairwise token similarities between ViFM and VDM representations to distill spatial constraints within frames and temporal dynamics across frames from the ViFM. Specifically, let $\mathbf{E_v}$ be the ViFM encoder processing video $\mathbf{V} \in \mathbb{R}^{F \times C \times H \times W}$. It outputs features $\mathbf{y_v} = E_v(\mathbf{V}) \in \mathbb{R}^{N \times D}$, where $N = f \times h \times w$ is the token count over $f$ temporal and $h \times w$ spatial positions, with feature dimension $D$. $(F/f, H/h, W/w)$ represent the temporal/spatial compression ratios. For VDMs, the 3D VAE encoder [56] compresses $\mathbf{V}$ into latent $\mathbf{z}$. The hidden state $\mathbf{h_t} = f_\theta(\mathbf{z_t})$ of denoising transformer is derived from noisy latent $\mathbf{z_t}$. The $\mathbf{h_t}$ is input into a trainable MLP $h_\phi$ for dimension $D$ alignment, i.e., $h_\phi(\mathbf{h_t}) \in \mathbb{R}^{f \times h \times w \times D}$. **Note:** Although the dimensions $(f, h, w)$ might differ from those derived from the ViFM output, we use the same notation here for annotation simplicity, representing the dimensions after ensuring compatibility.

We compute spatial token pairwise similarity matrix first. After reshaping $\mathbf{y_v}$ to $\mathbb{R}^{f \times (hw) \times D}$, the relation (i.e., cosine similarity) matrix for spatial dimension at frame $d$ can be expressed as:

$$y_{\text{spatial}}^{d,i,j} = \frac{\mathbf{y_v^{d,i}} \cdot \mathbf{y_v^{d,j}}}{\|\mathbf{y_v^{d,i}}\|\|\mathbf{y_v^{d,j}}\|} \tag{3}$$

where $i, j \in [1, hw]$ index spatial positions. This produces $\mathbf{y_{\text{spatial}}^d} \in \mathbb{R}^{hw \times hw}$ per frame. Aggregating across $f$ frames yields $\mathbf{y_{\text{spatial}}} \in \mathbb{R}^{f \times hw \times hw}$. Features are normalized before computing similarity.

Then for temporal relation, we compute cross-frame similarities between each token in frame $d$ and all tokens from other frames $e \neq d$. Let $\mathbf{y_v} \in \mathbb{R}^{f \times (hw) \times D}$ be reshaped foundation model features. For each frame $d$ and token position $i$:

$$y_{\text{temp}}^{d,i,j,e} = \frac{\mathbf{y_v^{d,i}} \cdot \mathbf{y_v^{e,j}}}{\|\mathbf{y_v^{d,i}}\|\|\mathbf{y_v^{e,j}}\|}, \quad \forall e \in [1, f] \setminus \{d\}, \ j \in [1, hw] \tag{4}$$

This produces a 4D tensor $\mathbf{y_{\text{temp}}} \in \mathbb{R}^{f \times hw \times hw \times (f-1)}$. Corresponding spatial similarity matrix $\mathbf{h_{\text{spatial}}}$ and temporal similarity matrix $\mathbf{h_{\text{temp}}}$ are computed identically using the VDM features $h_\phi(\mathbf{h_t})$.

The TRD loss then quantifies the difference between VDM and ViFM by calculating the average L1 distance using the corresponding spatial and temporal similarity values

$$\mathcal{L}_{\text{TRD}} = \frac{1}{f^2(hw)^2} \left( \underbrace{\sum_{d=1}^{f} \sum_{i,j=1}^{hw} \left| \mathbf{h_{\text{spatial}}^{d,i,j}} - \mathbf{y_{\text{spatial}}^{d,i,j}} \right|}_{\text{Spatial component}} + \underbrace{\sum_{\substack{d,e=1 \\ e \neq d}}^{f} \sum_{i,j=1}^{hw} \left| \mathbf{h_{\text{temp}}^{d,i,j,e}} - \mathbf{y_{\text{temp}}^{d,i,j,e}} \right|}_{\text{Temporal component}} \right) \tag{5}$$

The final loss can be expressed as $\mathcal{L} = \mathcal{L}_{\text{diff}} + \lambda \mathcal{L}_{\text{TRD}}$ where $\lambda$ is a hyperparameter. The TRD loss can be computed in a unified operation that, given a frame, directly calculates its self-similarity and its similarity with other frames, as implemented in the code.

### 3.4 Remaining issues for implementation

Successfully applying the Token Relation Distillation (TRD) loss, as detailed in Section 3.3, requires addressing practical implementation challenges, primarily concerning feature dimensionality and input configuration for Video Foundation Models (ViFMs).

Table 1: Results of Videophy. † denotes the results reported from WISA [46] and ∗ denotes detailed prompt input, see Section 4.2. Semantic Adherence (SA) measures the video-text alignment and fidelity. **Importantly**, Physical Commensense (PC) measures whether generated videos follow the physics laws in the real-world.

| Methods | Solid-Solid | | Solid-Fluid | | Fluid-Fluid | | Overall | |
|---|---|---|---|---|---|---|---|---|
| | SA | PC | SA | PC | SA | PC | SA | PC |
| VideoCrafter2 | 50.4 | 32.2 | 50.7 | 27.4 | 48.1 | 29.1 | 50.3 | 29.7 |
| DreamMachine | 55.1 | 21.7 | 59.6 | 23.3 | 58.2 | 18.2 | 57.5 | 21.8 |
| LaVIE | 40.8 | 18.3 | 48.6 | 37.0 | 69.1 | 50.9 | 48.7 | 31.5 |
| Cosmos-Diffusion-7B† | - | - | - | - | - | - | 57 | 18 |
| HunyuanVideo∗ | 55.2 | 16.1 | 67.1 | 30.1 | 54.5 | 54.5 | 60.2 | 28.2 |
| PhyT2V† | - | - | - | - | - | - | 61 | 37 |
| WISA (Koala dataset)† | - | - | - | - | - | - | 62 | 33 |
| WISA (WISA dataset)† | - | - | - | - | - | - | 67 | 38 |
| CogVideoX-2B | 37.8 | 12.6 | 67.1 | 30.1 | 45.5 | 50.9 | 51.6 | 26.2 |
| CogVideoX-2B∗ | 49.6 | 13.3 | 71.2 | 28.1 | 60.0 | 50.9 | 60.5 | 25.6 |
| **VideoREPA-2B**∗ | 52.4 | 18.2 | 77.4 | 32.2 | 60.0 | 52.7 | 64.2 | 29.7 |
| CogVideoX-5B† | - | - | - | - | - | - | 60 | 33 |
| CogVideoX-5B | 53.1 | 18.2 | 75.3 | 32.9 | 56.4 | 61.8 | 63.1 | 31.4 |
| CogVideoX-5B∗ | **62.9** | 19.6 | 76.0 | 33.6 | 72.7 | 61.8 | 70.0 | 32.3 |
| **VideoREPA-5B**∗ | 58.0 | **28.0** | 82.9 | **39.0** | 80.0 | **74.5** | 72.1 | **40.1** |

A key issue is the dimensional misalignment between features of ViFM and VDM. After their respective encoding processes, the temporal $f$ and spatial $h \times w$ dimensions often differ. Advanced VDMs [56, 45, 24] frequently employ 3D VAEs with high temporal compression, e.g., 4x or 8x. In contrast, ViFMs [47, 5] typically use lower compression ratios, e.g., 2x. This results in ViFM feature maps $\mathbf{y_v}$ having a larger temporal size, and often different spatial sizes, compared to VDM latents $\mathbf{h_t}$. To reconcile these differences while maximizing the guidance from the ViFM, we adopt the principle of interpolating VDM latent dimensions to match ViFM features, a strategy empirically found to be more effective. Another consideration arises from computational resource limitations when processing inputs for ViFMs, which often utilize 3D full attention. Inputting high-resolution video, e.g., 480x720, as in CogVideoX or a large number of frames, e.g., 49 frames, as in CogVideoX directly into a ViFM can be prohibitively memory-intensive. This necessitates a trade-off. We explore three strategies and finally decide to process all frames at a lower resolution to preserve the integrity. Experiments, as shown in Appendix D, are conducted to support this decision/approach.

## 4 Experiments

### 4.1 Implementation details

**Model setups.** We adopt CogVideoX [56], a powerful T2V diffusion model, as the base model and fintune it using the proposed TRD loss. Specifically, we develop VideoREPA-2B and VideoREPA-5B, corresponding to the ones in CogVideoX. The generated videos consist of 49 frames at a resolution of 480×720. We adopt VideoMAEv2 [47] as the alignment target encoder.

**Training details.** Unlike methods such as WISA [46] that require specialized datasets with explicit physical phenomena (e.g., WISA-32K [46]), we leverage OpenVid [35], a large-scale, open-domain video dataset. Videos are center-cropped and resized to 480×720. VideoREPA-2B is finetuned on 32k OpenVid videos for 4,000 steps. For the larger VideoREPA-5B, we use LoRA for efficient finetuning on 64k OpenVid videos for 2,000 steps. The default alignment depth is 18. All experiments utilize 8 NVIDIA A100 (80GB) with a total batch size of 32. More details are shown in Appendix B.

### 4.2 Evaluation

We evaluate VideoREPA on two challenging benchmarks designed to comprehensively assess the physical plausibility of videos generated by text-to-video models:

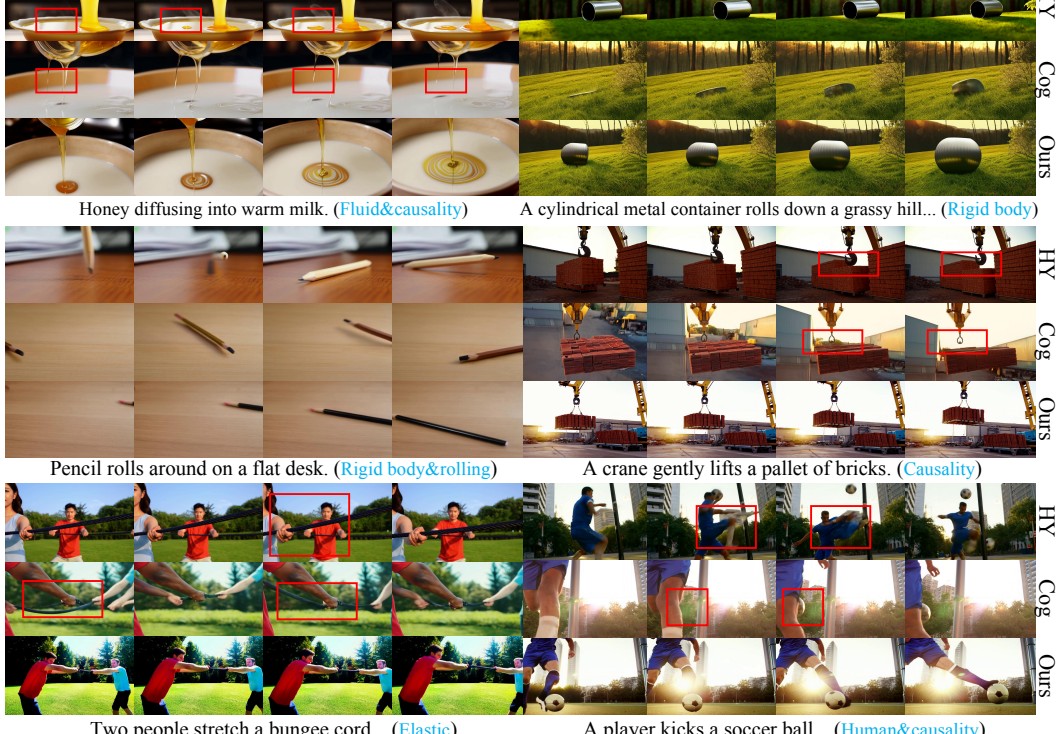

HY

Cog

Ours

Honey diffusing into warm milk. (Fluid&causality)

A cylindrical metal container rolls down a grassy hill... (Rigid body)

Pencil rolls around on a flat desk. (Rigid body&rolling)

A crane gently lifts a pallet of bricks. (Causality)

Two people stretch a bungee cord... (Elastic)

A player kicks a soccer ball... (Human&causality)

Figure 3: Qualitative comparison of HunyuanVideo (HY)[24], CogVideoX (Cog)[56], and Video-REPA (Ours), exhibiting enhanced physics commonsense of VideoREPA.

**VideoPhy [3].** This benchmark uses 344 prompts to test if generated videos adhere to physical commonsense in real-world activities, covering diverse material interactions (e.g., solid-solid, solid-fluid) to evaluate whether VDMs can generate videos with plausible physics. The proposed VideoCon-Physics [3] is adopted as auto-rater in our paper. When testing CogVideoX and our VideoREPA, we refine the short prompt into a detailed prompt. This is crucial because the models are trained with long prompts, and prompts impact the quality of the video generation according to the official of CogVideoX [56]. Details are shown in Appendix E. Following WISA [46], we set SA = 1 and PC = 1 when their values are greater than or equal to 0.5. Values less than 0.5 are set as SA = 0 and PC = 0.

**VideoPhy2 [4].** VideoPhy2 is an action-centric benchmark designed to evaluate physical common-sense in generated videos. It aims to overcome limitations of prior benchmarks, such as restricted size, absence of human interaction, and sim-to-real discrepancies. The dataset includes 590 detailed testing prompts covering 200 diverse actions. For automated evaluation, we employ the VideoPhy2-AutoEval model. We utilize the upsampled prompts provided by the VideoPhy2 benchmark.

### 4.3 Quantitative comparisons

Using auto-evaluators for VideoPhy and VideoPhy2, we quantitatively assess our VideoREPA. To show its superiority, VideoREPA is benchmarked against its CogVideoX [56] baseline, other leading T2V models (VideoCrafter2 [9], LaVie [50], DreamMachine [30], Cosmos-Diffusion [1], HunyuanVideo [24]), and physics-aware methods like PhyT2V [55] and WISA [46].

Results in Table 1 show VideoREPA achieves state-of-the-art performance across three interaction types. Compared to its baseline CogVideoX, **VideoREPA-5B improves the Physical Commonsense (PC) score by 24.1% overall (specifically, 42.9% for Solid-Solid, 16.7% for Solid-Fluid, and 20.6% for Fluid-Fluid)**. Our method also surpasses WISA [46], a technique designed for enhancing physics commonsense in video generation. Notably, while WISA shows efficacy when trained on the physics-explicit dataset WISA-32K [46], it struggles to generalize to open-domain datasets like

Koala-36M [48]. In contrast, VideoREPA, trained on an open-domain dataset, demonstrates clear improvements over WISA on such data (e.g., PC score of 40.1 vs. WISA's 33 on Koala-36M).

We further assess physical commonsense on VideoPhy2 [4], an action-centric benchmark featuring complex human-object interactions. Following its protocol, Semantic Adherence (SA) and Physical Commonsense (PC) scores are the proportion of videos rated $\geq 4$ for each metric. Our VideoREPA (2B) demonstrates a significant improvement over the baseline by **4.57 scores** as shown in Table 2, further validating the effectiveness of our proposed method.

### 4.4 Qualitative comparisons

We present qualitative comparisons of videos generated by different models in Figure 3. Our VideoREPA achieves superior physics plausibility compared to HunyuanVideo and CogVideoX. Specifically, in the "pencil roll" scenario, videos from HunyuanVideo and CogVideoX often depict pencils rolling in a manner inconsistent with rigid body

Table 2: Results of Videophy2

| Methods | SA | PC |
|---|---|---|
| CogVideoX | **21.02** | 67.97 |
| VideoREPA | **21.02** | **72.54** |

motion laws. In contrast, VideoREPA showcases physically consistent and stable motion. Similarly, for the "crane lifting bricks" example, VideoREPA accurately portrays the crane maintaining a physical connection while lifting the pallet. The other methods, however, tend to generate videos where the bricks are implausibly suspended without any visible means of support from the crane.

For clarity, all prompts are shown in the short version, but the models received the detailed. More results are provided in Appendix G. Check detailed prompts and videos at Project Page.

### 4.5 Ablation studies

We conduct ablation studies to reveal the properties and validate the effectiveness of our proposed VideoREPA. Performance of VideoREPA-2B on VideoPhy is reported unless otherwise specified.

Table 3: Ablation study on TRD loss. NaN means only $\mathcal{L}_{\text{diff}}$ is adopted.

| Loss Type | SA | PC |
|---|---|---|
| NaN | 63.6 | 23.2 |
| TRD loss | **64.2** | **29.7** |
| only spatial | 61.0 | 27.3 |
| only temporal | 61.0 | 27.9 |

**Token Relation Distillation loss.** We ablate on TRD loss to assess its effectiveness. Physical plausibility relies on correct spatial appearance (e.g., no irregular deformations) and coherent temporal dynamics (e.g., smooth, accurate motion). We design TRD loss with both spatial and temporal terms to address these. The PC scores in Table 3 confirm their importance, showing that removing either component degrades performance. Interestingly, focusing alignment on only the spatial or temporal dimension negatively impacts Semantic Adherence (SA), likely due to harming the integrity of learned representation of VDMs.

**The ineffectiveness of REPA.** As discussed in Section 3.3, directly applying REPA [58] for physics enhancement via finetuning pre-trained VDMs presents several challenges. Results in Figure 4 demonstrate this: finetuning a VDM with the standard REPA loss leads to a significant degradation in video semantic quality. This outcome supports our assertion that REPA, with its "hard" alignment approach (i.e., token similarity), is unsuitable for finetuning pre-trained VDMs as it can disrupt their established feature spaces. In contrast, our proposed TRD loss,

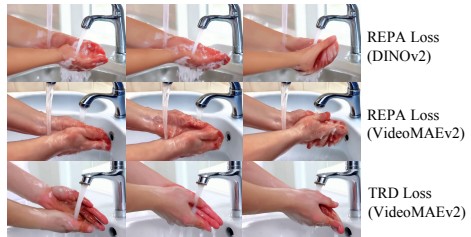

Figure 4: Ablation on REPA loss.

which offers "soft" guidance, proves substantially more effective for finetuning VDMs.

## 5 Conclusion and outlook

In this paper, we presented VideoREPA, a framework designed to transfer physics knowledge from Video Foundation Models (ViFMs) to text-to-video diffusion models (VDMs) via token-level relation distillation. We first identified a significant physics understanding performance gap between ViFMs and VDMs. Subsequently, motivated by the principle that enhanced understanding facilitates

higher-quality generation, we proposed the Token Relation Distillation (TRD) loss to distill physics understanding capability from pre-trained ViFMs to VDMs, thereby achieving more physically plausible video generation. Extensive experiments demonstrate that VideoREPA achieves state-of-the-art generation results, exhibiting great physical commonsense in generated videos.

**Limitations.** Although VideoREPA has achieved significant improvement through fine-tuning VDMs, its potential for pre-training VDMs remains unvalidated due to computational resource limitations. Future research could explore incorporating VideoREPA into the pre-training of VDMs and developing targeted innovations to effectively inject physics knowledge during this phase.

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

# Appendix

## A    Analysis on why TRD loss contribute to physics knowledge transfer

A key objective of our framework is to enhance the physical plausibility of video generation by transferring physics-related knowledge from video foundation models (ViFMs) to video diffusion models (VDMs). Existing VDMs exhibit limited capability in modeling physical dynamics, whereas ViFMs have demonstrated strong emergent understanding of object configurations and motion causality. Distilling relational representations from ViFMs into VDMs therefore serves as an effective mechanism to bridge this capability gap. Rather than being a byproduct of training, physical consistency is an explicit optimization objective in our framework, and the Token Relation Distillation (TRD) loss is specifically formulated to encode this objective.

The TRD loss operates by aligning spatial and temporal relational matrices between the ViFM and the VDM. The spatial component enforces consistency in static structural relations such as object integrity, contact relationships, and positional dependencies within each frame. The temporal component captures dynamic interactions, including motion trajectories, causality, and inter-frame dependencies, which are essential for modeling intuitive physics. By aligning these relational structures instead of matching raw features, TRD induces the VDM to internalize physical rules governing object interactions and temporal evolution. Empirical results (see Figure 3) demonstrate that VDMs trained with TRD exhibit improved physics understanding in intermediate feature evaluations and generate videos with significantly enhanced physical plausibility (Tables 1 and 2), validating the effectiveness of relational knowledge transfer.

## B    Detailed training setting

For finetuning, we utilize the OpenVid dataset [35], an open-source, high-quality collection of videos with expressive captions, containing over one million in-the-wild videos. The learning rate is set to 1e-4 for LoRA-based finetuning of CogVideoX-5B and 2e-6 for full-parameter finetuning of CogVideoX-2B. For LoRA, the rank is 128 and alpha is 64. The target encoders explored in our experiments include VideoMAEv2-B [47], V-JEPA-L [5], OmniMAE-B [18], and VideoMAE-B [43]. Unless otherwise specified, an alignment depth of 18 is used for both VideoREPA-2B and VideoREPA-5B. Inspired by VA-VAE [57], to prevent the alignment of unnecessary noise, we incorporate a margin $m$ (typically ranging from 0 to 0.1) into the TRD loss (Equation (5)). Specifically, values in TRD loss less than this margin are set to 0. The appropriate margin value was found to vary: 0.1 for VideoMAEv2, 0.05 for V-JEPA, and 0 for both VideoMAE and OmniMAE, reflecting the great fit for each encoder.

## C    Physion evaluation setting

For the physics understanding evaluation discussed in Section 3.2, we utilize the Physion benchmark [6]. Physion presents realistic simulations of diverse physical scenarios, where objects are manipulated in various configurations to assess different types of physical reasoning, including stability, rolling motion, and object linkage, among others. We specifically employ Physion v1.5 [6], the latest version, which features improved rendering quality and more physically plausible simulations.

Physion stands out as a challenging benchmark due to its inclusion of diverse physical phenomena, complex object dynamics, and realistic 3D simulations. These characteristics make it a preferable choice over other benchmarks like ShapeStacks [20] and IntPhys [39], which offer comparatively limited object dynamics.

Specifically, for feature extraction from CogVideoX and VideoREPA, we select features from three temporal dimensions, evenly sampled from the twelve available temporal dimensions in their respective latent spaces. All spatial tokens within these selected temporal slices are utilized. We employ the Object Contact Prediction (OCP) task from the Physion for evaluation. The OCP task assesses a model's capability to predict future contact between two objects based on an initial context video, requiring an implicit understanding of physical dynamics for accurate prediction.

The evaluation procedure involves first extracting features using the VDM. Consistent with our alignment strategy, we extract these features from the 18th layer of the denoising network. Subsequently,

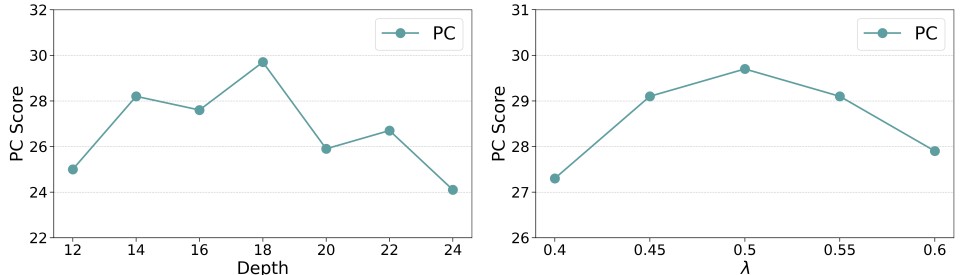

Figure 5: *Left:* Effect of alignment depth. *Right:* Effect of $\lambda$. PC score on VideoPhy is reported.

Table 4: Ablation study on different alignment target video foundation models.

| Models | SA | PC |
|---|---|---|
| - | 63.6 | 23.2 |
| VideoMAE [43] | 59.8 | 26.2 |
| V-JEPA [5] | 64.5 | 24.7 |
| OminiMAE [18] | 61.6 | 24.7 |
| V-JEPA 2 [2] | **65.1** | 27.3 |
| VideoMAEv2 [47] | 64.2 | **29.7** |

these extracted features are used to train a logistic regression classifier to perform the OCP task, i.e., predicting future object contact. For this evaluation, we utilize the "roll" and "contain" subsets of the Physion benchmark, with prediction accuracies reported in Figure 1.

## D    Additional ablation study

Additional ablation studies are conducted, aligning the experiment setting with Section 4.5.

**Different video foundation models.** We evaluate aligning the VDM with various pre-trained ViFMs: VideoMAE [43], V-JEPA [5], V-JEPA 2 [2], and VideoMAEv2 [47]. Results in Table 4 indicate VideoMAEv2 performs best, likely due to its extensive pre-training on millions of videos and resulting strong generalization. Thus we choose to align VDM with VideoMAEv2 in our VideoREPA.

**Different alignment depth.** We also investigate the effect of aligning different layers of the diffusion models with features extracted from the ViFM. Experiments in Figure 5 indicate that an alignment depth of 18 yields the best performance, which is adopted in our VideoREPA.

**Effect of $\lambda$.** We try different values of $\lambda$ for the weight of TRD loss. The Figure 5 shows that the $\lambda = 0.5$ features the best trade-off between the original diffusion loss and the proposed TRD loss.

**Dimension alignment.** We conduct an ablation study on the dimension alignment issue when applying the TRD loss. The temporal dimension of VideoREPA's latent space is typically smaller than that of VideoMAEv2's features, while its spatial dimensions are often larger. Our guiding principle is that interpolating VDM representations (from VideoREPA) to match the dimensions of ViFM features (from VideoMAEv2) best preserves the ViFM's knowledge. The experiments in Table 5 support this choice. Thus, we interpolate the VDM's latent representations to match the feature dimensions of the pre-trained ViFM. Furthermore, considering that the first encoded frame in the latent space of 3D VAE primarily serves to maintain semantic information [56], we exclude it from the alignment process to focus on dynamic content.

**Trade off between input frames and resolution.** Given the computational expense of full attention mechanisms in ViFMs, directly inputting high-resolution video, e.g., 480x720, as used by CogVideoX for generation or a large number of frames, e.g., 49 frames into a ViFM can be prohibitively memory-intensive. This necessitates a careful trade-off between input frame count and resolution for ViFM processing. We explored three common strategies to manage this:

1. Processing all video frames at a uniformly lower resolution.

Table 5: Dimension alignment target. Target dimension is VDM refers to interpolating ViFM features to match VDM dimensions.

| Spatial | Temporal | SA | PC |
|---------|----------|------|------|
| VDM | VDM | 63.6 | 26.2 |
| ViFM | VDM | 63.1 | 28.5 |
| ViFM | ViFM | **64.2** | **29.7** |

Table 6: Trade-off between resolution and frames. Corresponding strategies related to indexes are stated in the Appendix D.

| Index | SA | PC |
|-------|------|------|
| 1 | **64.2** | 29.7 |
| 2 | 63.4 | 29.1 |
| 3 | 54.7 | **32.0** |

2. Processing temporally grouped subsets of frames at high resolution.

3. Processing all frames at high resolution but with spatial cropping into patches or groups.

Based on empirical evaluations in Table 6, we adopted the first strategy: processing all frames at a reduced resolution. This approach was found to best preserve the holistic nature of the ViFM's pre-trained representations with the lowest computation resources needed, whereas the latter two strategies (grouping or cropping) tended to degrade either physics plausibility or semantic quality of the generated videos.

## E  Generating detailed prompt for VideoPhy

Adhering to guidance from the official CogVideoX documentation [56], which emphasizes the criticality of refining prompts, we specifically elaborate the often-brief prompts found in the VideoPhy benchmark [3]. Poorly formulated prompts can significantly degrade Semantic Adherence, consequently hindering validations of the perceived Physical Commonsense. To mitigate ambiguity, we leverage Large Language Models (LLMs) such as GPT-4o or Gemini 2.5 Pro. These LLMs are tasked with clarifying vague expressions and explicating implicit details within the original prompts, thereby minimizing confounding factors. We detail the prompts from VideoPhy using the following template:

> You are a reasoning expert. Your task is to examine the given user prompt and identify whether there is any implicit knowledge that should be made explicit in the description. Your goal is to refine the prompt by making all details clear and descriptive, ensuring that no reasoning is required for understanding the outcome, environment, or processes involved. This means removing any assumptions or implicit components, such as environmental context, sequence of actions, or the cause-and-effect process, that are not immediately obvious.
>
> Please rewrite the original prompt in a clear, descriptive manner, without including any formulas or unnecessary reasoning, while providing as much detail as possible about the scene, actions, and effects. You should create a polished version of the prompt where the outcome is immediately clear to the reader, leaving no room for ambiguity. Some in-context examples are provided for your reference, and you need to finish the current task: ...
>
> Original prompt: A blender spins, mixing squeezed juice within it.
>
> Let's think step by step. The refined prompt should be less than 150 words.

## F  User study

While quantitative metrics provide useful indicators of video generation performance, they may not fully capture human-perceived physical plausibility. To complement the quantitative evaluation and directly assess perceptual realism, we conducted a user study using the Good-Same-Bad (GSB) pairwise comparison protocol [14]. Participants were shown videos generated by **VideoREPA-5B** and **CogVideoX-5B** (the finetuning baseline trained on the same dataset using only $\mathcal{L}_{\text{diff}}$) for 344 prompts from the VideoPhy dataset. Each video pair was evaluated along two axes: (*i*) **Semantic Adherence (SA)**, measuring the alignment between the generated video and the input text prompt; and (*ii*) **Physical Commonsense (PC)**, assessing the plausibility of physical dynamics such as object interactions, motion continuity, and causal consistency.

Table 7: User study results comparing VideoREPA and CogVideoX with respect to Semantic Adherence (SA) and Physical Commonsense (PC). Values indicate the percentage of user preference.

| Criterion | VideoREPA Wins | CogVideoX Wins | Tie |
|---|---|---|---|
| Semantic Adherence (SA) | **6.4** | 3.8 | 89.8 |
| Physical Commonsense (PC) | **21.7** | 8.0 | 70.3 |

As summarized in Table 7, the two models perform comparably in semantic adherence and Video-REPA demonstrates a substantial improvement in physical plausibility, being preferred 21.7% of the time compared to 8.0% for the baseline, representing a $2.7\times$ relative gain. These results confirm that VideoREPA effectively captures human-perceived physical commonsense while maintaining semantic fidelity.

## G   More qualitative results

In this section, we present additional video generation results from our VideoREPA-5B model, along with outputs from the baseline CogVideoX-5B. This comparison aims to further demonstrate the enhanced physical commonsense achieved by our method in the generated videos. Figures 6 to 8 illustrate these direct comparisons, showcasing the superiority of VideoREPA. Additionally, to highlight its capabilities further, we display more videos generated by VideoREPA that exhibit strong physical plausibility in Figure 9 and Figure 10. Red rectangles denote phenomena that violate physical commonsense for easier distinguish.

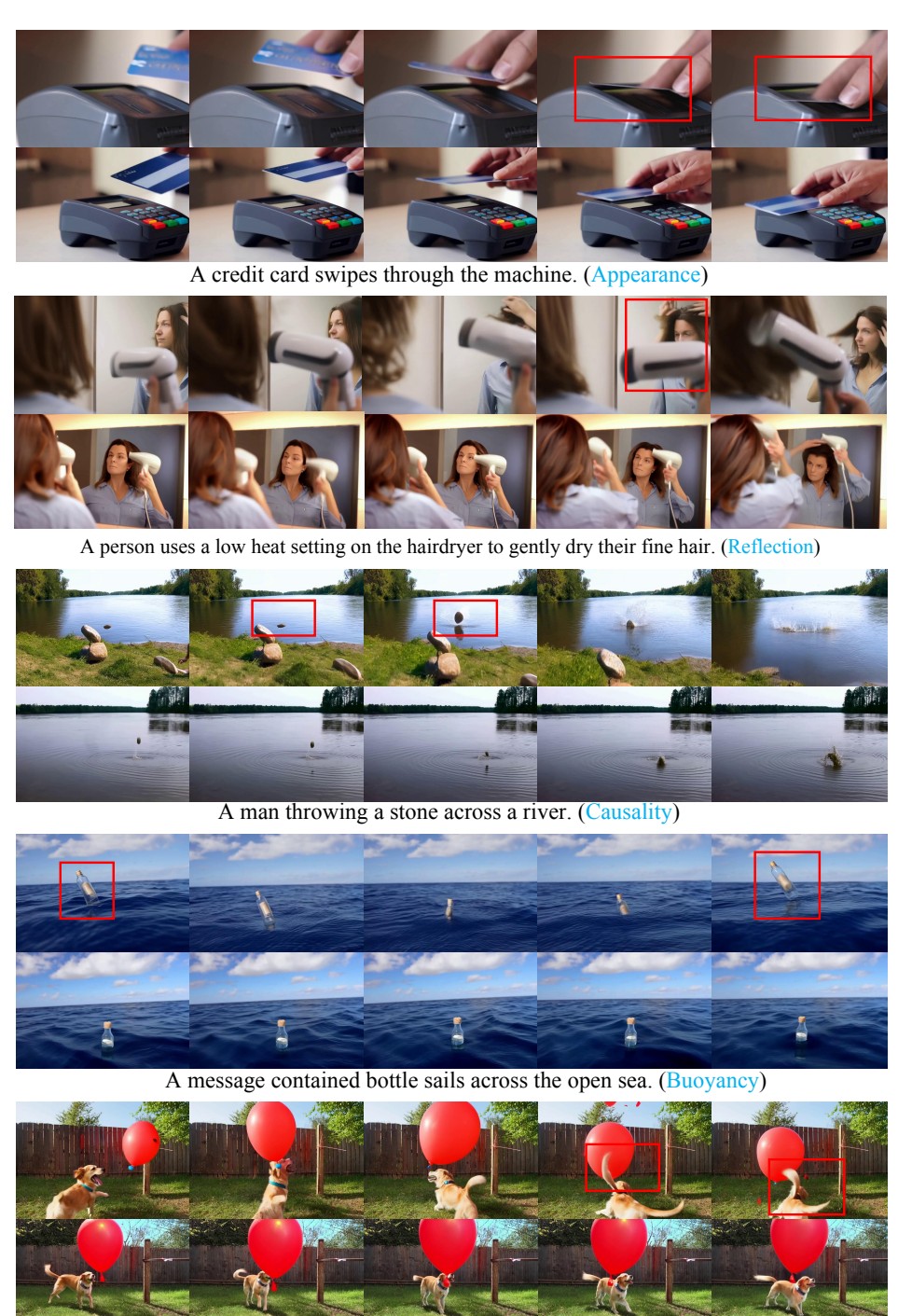

A credit card swipes through the machine. (Appearance)

A person uses a low heat setting on the hairdryer to gently dry their fine hair. (Reflection)

A man throwing a stone across a river. (Causality)

A message contained bottle sails across the open sea. (Buoyancy)

A dog playfully bats at a balloon... (Appearance)

Figure 6: Qualitative results. The first row displays the outcomes of CogVideoX, and the second row presents the results of our VideoREPA.

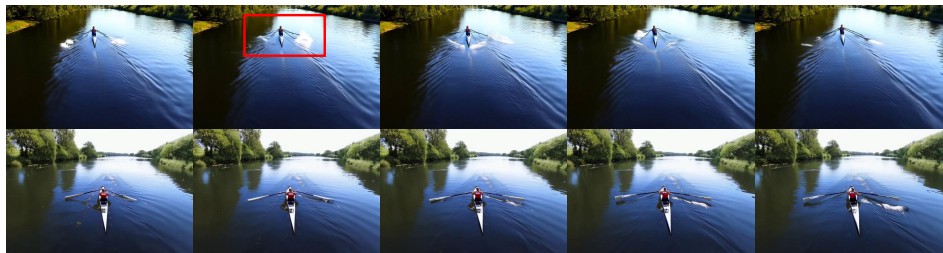

A single scull rower uses one oar to propel a boat. (Commonsense)

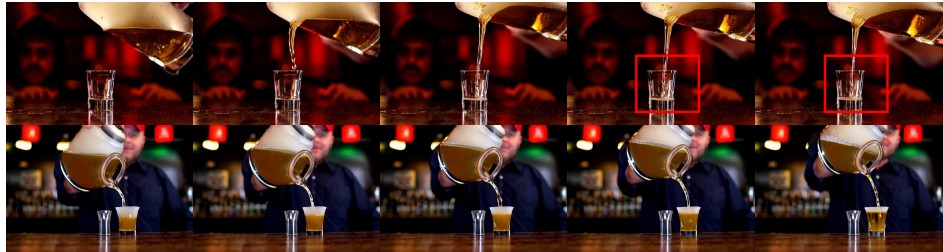

A person... pouring the remaining beer into a waiting shot glass. (Fluid)

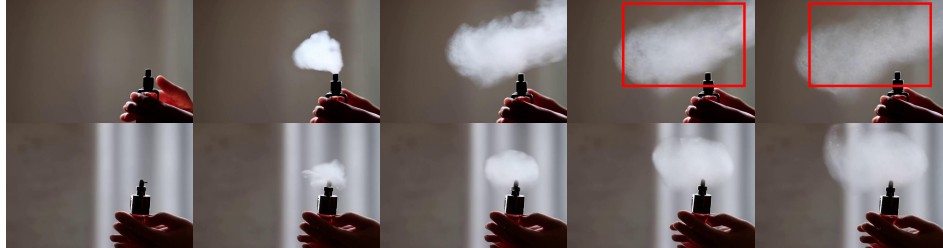

Perfume spraying from a perfume bottle. (Fluid&gas)

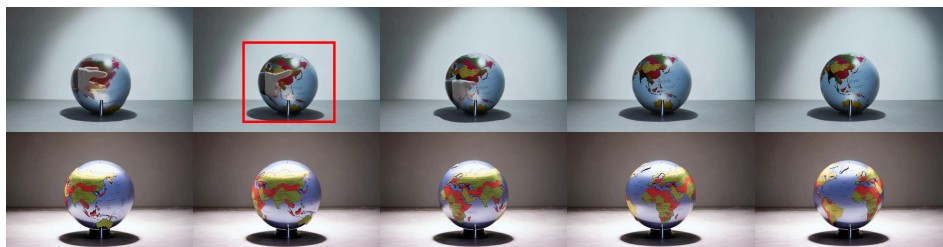

...a globe is poked, causing it to spin on its axis (Rigid body&rotation)

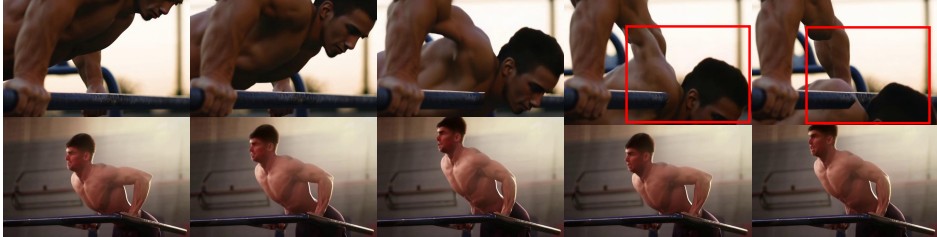

Parallel bars are shown from a side view with an athlete performing dips. (Commonsense)

Figure 7: Qualitative results. The first row displays the outcomes of CogVideoX, and the second row presents the results of our VideoREPA.

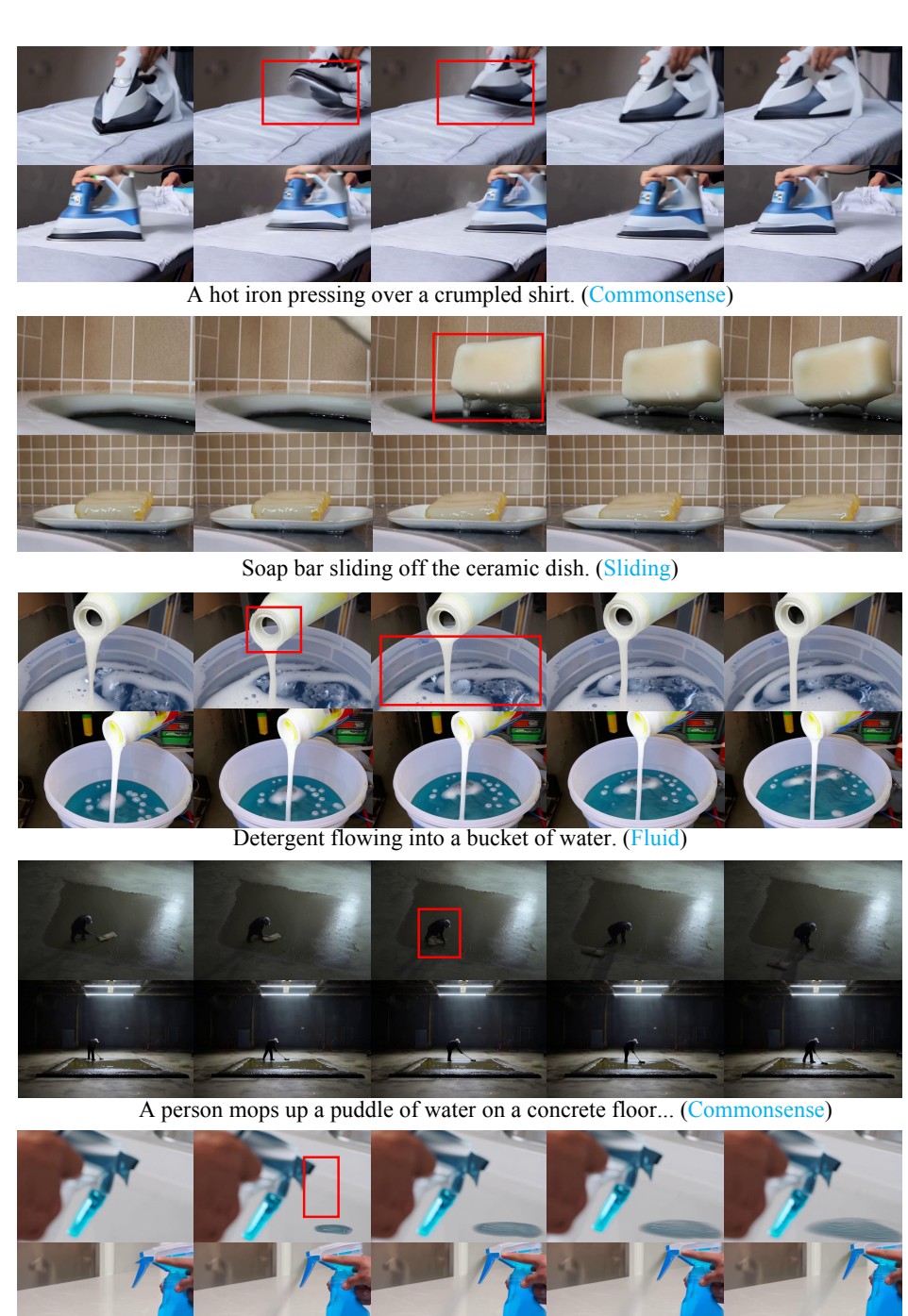

A hot iron pressing over a crumpled shirt. (Commonsense)

Soap bar sliding off the ceramic dish. (Sliding)

Detergent flowing into a bucket of water. (Fluid)

A person mops up a puddle of water on a concrete floor... (Commonsense)

A spray bottle sprays cleaning solution onto a countertop. (Causality)

Figure 8: Qualitative results. The first row displays the outcomes of CogVideoX, and the second row presents the results of our VideoREPA.

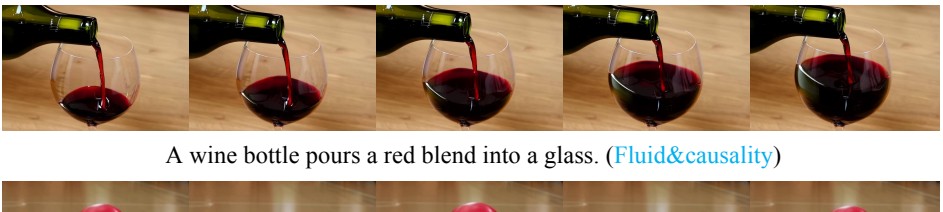

A wine bottle pours a red blend into a glass. (Fluid&causality)

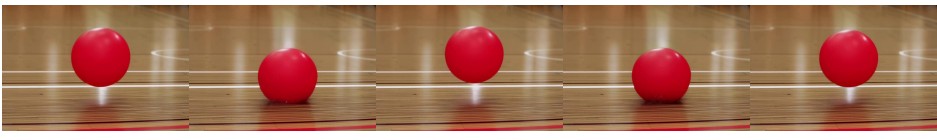

Ball bounces off the floor. (Gravity)

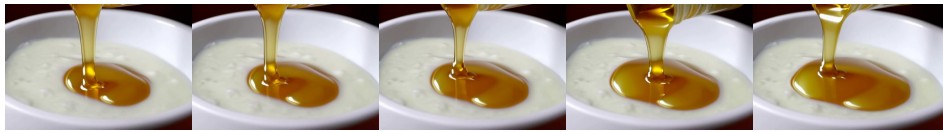

Drops of honey on smooth yogurt. (Causality)

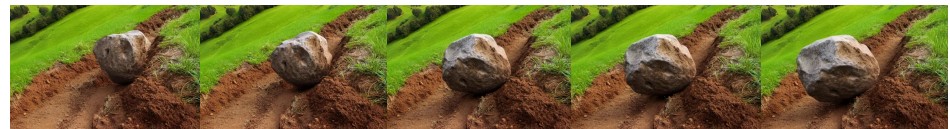

large stone rolls down a hillside... (Rigid body&roll)

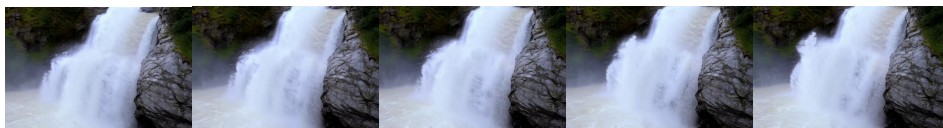

A waterfall cascades over jagged rocks... (Fluid)

Figure 9: Qualitative results, displaying videos generated by our VideoREPA.

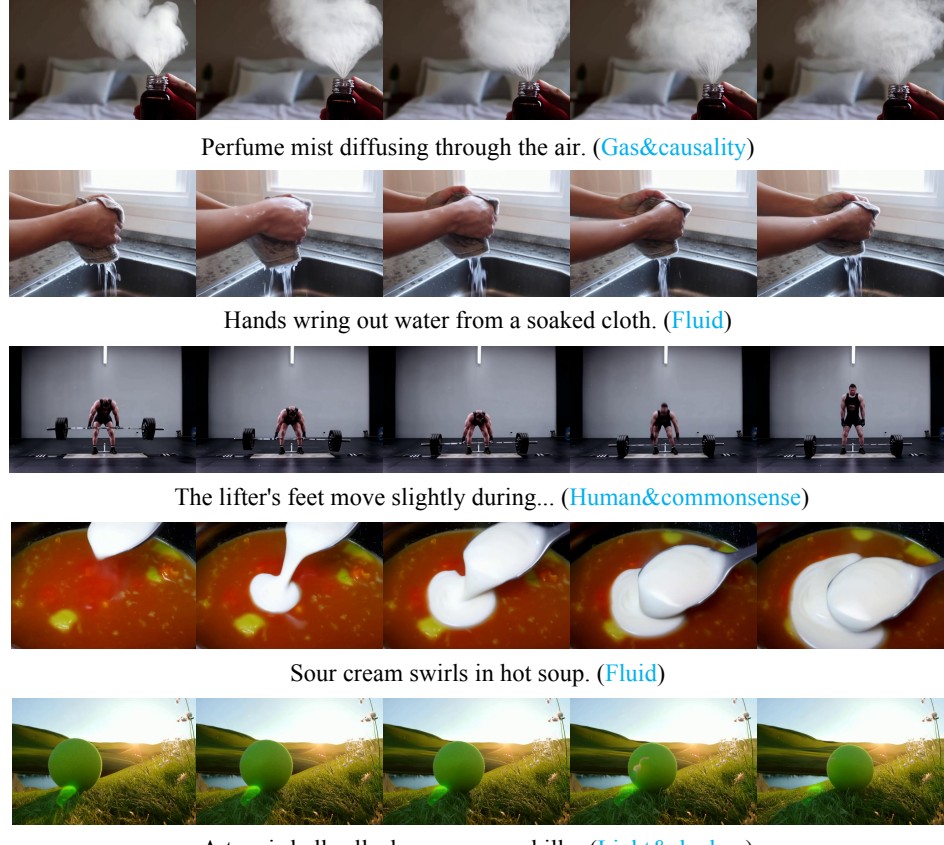

Perfume mist diffusing through the air. (Gas&causality)

Hands wring out water from a soaked cloth. (Fluid)

The lifter's feet move slightly during... (Human&commonsense)

Sour cream swirls in hot soup. (Fluid)

A tennis ball rolls down a grassy hill... (Light&shadow)

Figure 10: Qualitative results, displaying videos generated by our VideoREPA.

