# OpenReview forum: "VideoREPA: Learning Physics for Video Generation through Relational Alignment with Foundation Models"
_NeurIPS.cc/2025/Conference — NeurIPS 2025 poster_

### Official Review · Reviewer_JMpo · 2025-07-02

**Clarity:** 2
**Significance:** 3
**Originality:** 2
**Rating:** 4
**Confidence:** 3

**Summary:**

This work focuses on ensuring that text-to-video (T2V) models generate physically plausible content. Authors identify that the main limitation of current models is that the representations within such models have reduced capacity for physics understanding in contrast to self-supervised methods that focus on representational learning. Thus, a method to distill physics understanding from such foundational models into T2V models. This method enables fine-tuning of T2V to inject physical knowledge. The authors perform experiments comparing their results with CogVideoX.

**Questions:**

Questions were included in the previous section.

**Ethical Concerns:**

["NO or VERY MINOR ethics concerns only"]

**Final Justification:**

The response is clear, concise and it addresses my 3 concerns, especially the first 2 ones. Thus, I im inclined to keep my score.

**Limitations:**

Yes

**Quality:**

3

**Strengths And Weaknesses:**

# Strengths
* The core idea proposed by the authors is interesting and sound. Authors propose to leverage the representation capabilities of foundational video models, e.g., DinoV2 or VideoMAE2 to align the representations used by text-to-video latent diffusion models and thus capture more realistic and consistent dynamics (e.g., consistent object shapes and physical plausibility of how objects move and interact with other objects). At the same time the authors devise a method to achieve this alignment by enforcing relation similarity between token embeddings. Thus the alignment is not done directly over the representations but over their relations. This is a softer condition useful for fine-tuning as opposed to REPA.
* Qualitative results look good and the authors selected a set of videos that clearly show the difference between the baselines and the proposed method results. Quantitative evaluation is aligned with these results.
* In general the presentation is good (but writing could be improved).

# Weaknesses

In order of importance:

**Vague Link Between SSL Encoders and Physics**
The method claims to “enhance physics” by distilling knowledge from self-supervised video encoders. However, SSL encoders (e.g., DINOv2, or VideoMAE2) are typically trained to capture general motion or appearance representations — not necessarily physical plausibility or constraints like dynamics, contact, or force interactions.

What evidence is there that SSL video encoders actually encode meaningful physics properties (e.g., Newtonian constraints, conservation laws, contact dynamics) rather than just correlation patterns or appearance flow? Also, how do the authors ensure that the distilled relations truly correspond to physically plausible relations rather than dataset-specific visual correlations?

**Ambiguity in ‘Pairwise Token Similarity’ = Physics**
The logic assumes that aligning pairwise token similarities (via TRD loss) transfers physics. But pairwise similarity can capture texture, background motion, or trivial frame correlations — it doesn’t inherently encode contact constraints, collision avoidance, or kinematic feasibility. Why should pairwise token similarities enforce physically realistic behavior? Could this encourage spurious correlations instead of actual dynamic consistency?

**Writing and Clarity**

The writing could be improved to enhance readability and ensure a smooth flow of information and transitions between ideas. For example, Lines 40–41: *“Yet, WISA struggles to generalize to open-domain data, showing improvements primarily when trained on videos with explicit physics.”* This sentence is confusing and makes the paper more difficult to follow. Similar unclear phrasing can be found throughout the manuscript. Clarifying these statements would significantly improve the overall quality and accessibility of the paper.

---

> ### Author Rebuttal · Authors · 2025-07-31
>
> Thank you very much for your careful review and the recognition of our novelty, performance and great presentation. We clarify your questions below.
>
> > Q1.1. **Vague Link Between SSL Encoders and Physics** The method claims to “enhance physics” by distilling knowledge from self-supervised video encoders. However, SSL encoders (e.g., DINOv2, or VideoMAE2) are typically trained to capture general motion or appearance representations — not necessarily physical plausibility or constraints like dynamics, contact, or force interactions. What evidence is there that SSL video encoders actually encode meaningful physics properties (e.g., Newtonian constraints, conservation laws, contact dynamics) rather than just correlation patterns or appearance flow?
>
> Thanks for your question. It touches upon the core motivation of our work. It is partially correct that VFMs are typically trained to capture general motion or appearance representations. Apart from these features, **recent studies [1, 2, 3] point out that SSL representations capture high-level features (e.g. physics knowledge) beyond low-level appearance or motion.** For example, the paper [3] provides extensive experiments to show that VFMs like V-JEPA and VideoMAEv2 learn representations that align with physical concepts like object permanence and trajectory prediction without any explicit physics supervision, achieving great intuitive understanding ability.
>
> **We also validate that VFMs indeed encode physics information by assessing their ability on the Physion dataset** (Figure 1), a benchmark for tasks like object dynamics and contact prediction. The prediction accuracy of VFM VideoMAEv2 exceeds 70%, which is significantly higher than the 50% chance level and GPT-4 V only achieves 53%. This level of performance would be unattainable if the models were learning only superficial correlation patterns or appearance flow, rather than a deeper understanding of intuitive physics.
>
> Thus while it is true that Video SSL encoders like VideoMAEv2 are not explicitly trained with physics constraints (i.e., not learning explicit physics laws like Newtonian law, value of gravity and etc.), there is compelling evidence that they learn a rich, implicit understanding of intuitive physics which makes it able to predict, detect object dynamics and etc.
>
> > Q1.2. Also, how do the authors ensure that the distilled relations truly correspond to physically plausible relations rather than dataset-specific visual correlations?
>
> We ensure and validate this from two aspects. **Theoretically**, the distilled relations induced by carefully designed TRD loss contain spatial and temporal component which correspond to exactly the physics knowledge contained in videos. Because real-world physics is fundamentally about relationships including how objects are spatially configured (sptial component) and how these configurations evolve dynamically over time (temporal component) (details in answer to Q2).
>
> **Experimentally**, as shown in Table 1, our VideoREPA method leads to a substantial +24.1% improvement in the overall PC score for CogVideoX-5B. Also, further user study shows the supuriority of our VideoREPA in terms of physical realism in generated videos (The details are shown in the answer to Rewiewer CNDL, Q1). If we were merely distilling dataset-specific visual correlations, it is highly unlikely that this would translate into a consistent and significant improvement on a diverse set of prompts designed to test fundamental physical principles (e.g., solid-solid interaction, fluid dynamics, causality).
>
> **In summary of the answer to Q1**, we are not claiming that Video SSL encoders solve explicit physics equations or encode precise physical property / value. We are arguing, with evidence from our own work and the existing works, that they learn a powerful, implicit model of intuitive physics. Our results show that distilling this emergent knowledge into T2V models is an effective strategy for enhancing their physical realism.
>
> > Q2. **Ambiguity in ‘Pairwise Token Similarity’ = Physics** The logic assumes that aligning pairwise token similarities (via TRD loss) transfers physics. But pairwise similarity can capture texture, background motion, or trivial frame correlations — it doesn’t inherently encode contact constraints, collision avoidance, or kinematic feasibility. Why should pairwise token similarities enforce physically realistic behavior? Could this encourage spurious correlations instead of actual dynamic consistency?
>
> Because we design TRD loss using spatial-temporal pair-wise similarity to capture physics knowledge intentionally for effective transfer. Real-world intutive physics is fundamentally about relationships including how objects are spatially configured and how these configurations evolve dynamically over time. Our TRD loss directly mirrors this structure:
>
> *   **Spatial Relation Alignment:** This component captures plausible object configurations, shape integrity, and static interactions (e.g., an object resting *on* a surface).
> *   **Temporal Relation Alignment:** This component captures the dynamics of motion, causality, and state changes (e.g., how an object moves from frame A to B).
>
> By aligning these relational matrices, we are not just matching features; we are forcing the VDM to learn the underlying **structural and dynamic rules** that govern a video, as understood by the VFM.
>
> Furthermore, **the empirical success of our method also provides the strongest evidence that we are distilling meaningful physics, but not spurious correlations**. As shown in Figure 1 (right), our method significantly closes the physics understanding gap between the T2V model and the VFM. This transferred understanding directly translates into better generation quality, evidenced by the substantial +24.1% improvement on the PC score in the VideoPhy benchmark (Table 1).
>
> > Q3. **Writing and Clarity** The writing could be improved to enhance readability and ensure a smooth flow of information and transitions between ideas. For example, Lines 40–41: “Yet, WISA struggles to generalize to open-domain data, showing improvements primarily when trained on videos with explicit physics.” This sentence is confusing and makes the paper more difficult to follow. Similar unclear phrasing can be found throughout the manuscript. Clarifying these statements would significantly improve the overall quality and accessibility of the paper.
>
> Thanks for your valuable suggestions regarding the writing. We apologize for any unclear phrasing and have carefully revised the manuscript to improve clarity and flow. Below are three examples of such improvements that will be reflected in the next version of our paper:
>
> 1. Lines 40-41:
>
>     Original: "Yet, WISA struggles to generalize to open-domain data, showing improvements primarily when trained on videos with explicit physics.".
>
>     Modified: "Yet, WISA only show improvement when training with dedicated collected dataset WISA-32K where each relates to individual explicit physical phenomena. It struggles to generalize to open-domain data which is much easily to collected and scale."
>
>     Comment on Improvement: We replaced the vague phrase "struggles to generalize" with a more concrete explanation of WISA's dependency on its specialized dataset (WISA-32K).
>
> 2. Lines 66-67:
>
>     Original: "thereby improving the physical realism of generated videos without relying on physics explicit datasets (e.g., WISA-32K)."
>
>     Modified: "thereby improving the physical realism of generated videos. Our method performs well on open-domain video dataset OpenVid without relying on the physics explicit dataset WISA-32K which is much harder to colloct."
>
>     Comment on Improvement: We restructured the sentence to first state our method's core benefit and then explicitly contrast its data requirements with WISA's.
>
> We made revisions throughout the paper to enhance its overall quality and accessibility. We are committed to further improving the text and welcome any additional feedback. Thank you again.
>
> [1] Yu S, Kwak S, Jang H, et al. Representation alignment for generation: Training diffusion transformers is easier than you think[J]. arXiv preprint arXiv:2410.06940, 2024.
>
> [2] Bordes F, Garrido Q, Kao J T, et al. IntPhys 2: Benchmarking Intuitive Physics Understanding In Complex Synthetic Environments[J]. arXiv preprint arXiv:2506.09849, 2025.
>
> [3] Garrido Q, Ballas N, Assran M, et al. Intuitive physics understanding emerges from self-supervised pretraining on natural videos[J]. arXiv preprint arXiv:2502.11831, 2025.

---

> ### Author Response · Authors · 2025-08-06
> **Looking forward to your further reply**
>
> Dear Reviewer JMpo,
>
> Thank you once again for your insightful and detailed review of our paper.
>
> We have provided a comprehensive rebuttal that directly addresses your concerns, including:
> 1.  **Evidence for SSL Encoders Learning Physics:** We presented evidence from both prior work and our own experiments (Fig. 1) to show that VFMs learn a rich, implicit understanding of intuitive physics.
> 2.  **Justification for TRD Loss:** We clarified how our TRD loss, by aligning spatio-temporal relations, is specifically designed to capture and transfer the relational nature of physical knowledge.
> 3.  **Writing and Clarity:** We have provided examples of revisions we will make to improve the paper's clarity, as you suggested.
>
> **As the discussion period is coming to an end**, we want to follow up and kindly ask if our responses have sufficiently addressed your questions. We would be very happy to respond to any further feedback you may have.
>
> Best regards,
>
> The Authors

---

> > ### Comment · Reviewer_JMpo · 2025-08-07
> >
> > Thanks to the authors for the response, it is clear, concise and it addresses my 3 concerns, especially the first 2 ones. Thus, I keep my score.

---

> > > ### Author Response · Authors · 2025-08-07
> > >
> > > Dear Reviewer JMpo,
> > >
> > > Thank you for your feedback and for confirming that our rebuttal has addressed your concerns. We appreciate your time and the constructive dialogue throughout this process.
> > >
> > > Your insightful questions have helped us improve the clarity and justification of our work. Please feel free to let us know if any further questions arise.
> > >
> > > Best regards,
> > >
> > > The Authors

---

### Official Review · Reviewer_ryaw · 2025-07-02

**Clarity:** 3
**Significance:** 3
**Originality:** 3
**Rating:** 4
**Confidence:** 5

**Summary:**

VideoREPA introduces an approach to enhance the physical plausibility of videos generated by T2V diffusion models. The method leverages an improved version of REPA by distilling physics knowledge from video foundation models into video diffusion models, focusing on finetuning rather than training from scratch.

**Questions:**

- In Figure 1 (right), the evaluation of physics understanding extracts intermediate features from VDMs to predict video futures (according to Appendix B). This approach raises concerns because VDMs are trained for denoising, not for extracting rich features like MAEs. My questions (or concerns) are:
1. Is it fair to compare **noisy activations** from VDMs with the structure features of MAEs?
2. Can you claim that large VDMs understand physics worse than smaller VFMs based on this unfair comparison (L47-50)?
3. Doesn't the enhanced alignment is a trivial results by directly aligning the feature with MAE, rather than the results of truly enhanced physics understanding?
- Why does the spatial component of the TRD loss not exclude the case $i=j$, while the temporal component does exclude $d=e$? Please clarify / justify this design choice.
- The quantitative results show VideoREPA-5B outperforming HunyuanVideo, yet visually the latter appears better in quality. Why was HunyuanVideo not used as the baseline model?

**Ethical Concerns:**

["NO or VERY MINOR ethics concerns only"]

**Final Justification:**

I justified the decision to increasing the score in the final comment.

**Limitations:**

The authors addressed the limitations of their work.

Common comment for the authors: If I couldn't fully understand your methods or am underestimating your contributions, feel free to explain and let's discuss. I will gladly raise the scores if I have misunderstood key points or if my concerns are addressed well.

I think the most critical points are:
- The first weakness described above.
- Validity of the comparison on Figure1 right (My first question).
- A broader quantitative comparison of video fidelity, consistency, and text-video alignment

**Quality:**

3

**Strengths And Weaknesses:**

### Strength
- The paper proposes a novel way to adapt REPA for finetuning, shifting its original scope from training acceleration to knowledge transfer.
- Experimental results and video supplements demonstrate that the physical plausibility of generated videos improves using the proposed method.
- The method effectively addresses the potential issue of feature dimension mismatch between VFMs and VDMs through interpolation.

### Weakness
- It remains unclear why the **physical knowledge** of VFMs should be distilled into VDMs as proposed. The manuscript gives the impression that enhanced physical plausibility is merely a training byproduct, not an explicit goal. The justification for why the TRD loss contributes to physics understanding needs to be clarified.
- The experiments are limited to CogVideoX-2B and 5B, which share the same structure and are now relatively outdated. Demonstrating the method's integration with more recent and diverse open-source video generation models would strengthen its credibility.
- A broader quantitative comparison of video fidelity, consistency, and text-video alignment (e.g. using VBench) is lacking. Without this, it is unclear whether the improvement in physical plausibility compromises other aspects of video quality.

---

> ### Author Rebuttal · Authors · 2025-07-31
>
> Thank you for the time, thorough comments, and questions. We appreciate your endorsement of our method in the aspects of novelty, performance and careful method design. Great to hear that you are willing to raise your score if concerns addressed and we'd be happy to clarify certain points for your.
>
> ### Weakness
>
> > Q1. It remains unclear why the physical knowledge of VFMs should be distilled into VDMs as proposed. The manuscript gives the impression that enhanced physical plausibility is merely a training byproduct, not an explicit goal. The justification for why the TRD loss contributes to physics understanding needs to be clarified.
>
> Thank you for this question. We want to emphasize that **enhancing physical plausibility is the explicit, primary goal of our framework, not a training byproduct.** Our entire research pipeline—from motivation to method design—is centered around this objective.
>
> **1. Motivation: Why Distill from VFMs to VDMs?**
>
> Our work stems from a clear, two-part observation: 1) (Problem) Poor Physics Plausibility and Understanding in VDMs. 2) (Opportunity) Strong Physics Understanding in VFMs.
>
> **Given that understanding generally benefits generation**, this significant physics understanding gap motivated our core idea: to bridge this gap by **distilling the emergent physical knowledge** from capable VFMs into powerful VDMs.
>
> **2. Method Design: Why TRD Loss Contributes to Physics Knowledge Transfer?**
>
> The design of our Token Relation Distillation (TRD) loss is tailored specifically to capture the essence of physical knowledge. Real-world intutive physics is fundamentally about relationships: how objects are spatially configured and how these configurations evolve dynamically over time. Our TRD loss directly mirrors this structure:
>
> *   **Spatial Relation Alignment:** This component enforces plausible object configurations, shape integrity, and static interactions (e.g., an object resting *on* a surface).
> *   **Temporal Relation Alignment:** This component captures the dynamics of motion, causality, and state changes (e.g., how an object moves from frame A to B).
>
> By aligning these relational matrices, we are not just matching features; we are forcing the VDM to learn the underlying **structural and dynamic rules** that govern a physical scene, as understood by the VFM.
>
> **3. Empirically Validating the Transfer of Physics Knowledge into VDMs**
>
> Our extensive experiments validate our claim in 2. The **physics understanding evaluation (Fig. 1)** confirms that TRD loss successfully transfers understanding capability. Subsequently, the **generation benchmarks (Table 1 & 2)** show this enhanced physics understanding induced by TRD loss translates into significantly improved physical plausibility in videos. This aligns with the established principle that understanding helps generation and **further shows that physical knowledge is indeed being distilled effectively.**
>
> > Q2. Demonstrating the method's integration with more recent and diverse open-source video generation models.
>
> Following your advice, we integrated VideoREPA with another open-source model: Wan2.1 (1.3B) [1]. The results on the VideoPhy:
>
> Models|SA|PC
> -|-|-
> Wan2.1|67.1|21.5
> VideoREPA (ours)|66.9|**26.7**
>
> **It shows that VideoREPA significantly improves upon the Wan2.1 in terms of PC (physical plausibility)**. This validates the generalization capability of our proposed method. Further visulizations will be updated in the next version of our paper.
>
> > Q3. A broader quantitative comparison of video fidelity, consistency, and text-video alignment (e.g. using VBench) is lacking.
>
> To verify that our improvement in physical plausibility does not compromise other aspects, we conducted experiments on VBench [2].
>
> The results, presented below, show that **VideoREPA is comparable to the CogVideoX across every dimension**. This provides strong evidence that our method successfully enhances physical plausibility without sacrificing general video fidelity, consistency, or text-video alignment.
>
> Model|Subject Consistency|Background Consistency|Temporal Flickering|Motion Smoothness |Dynamic Degree|Aesthetic Quality|Imaging Quality|Quality score (Overall)
> -|-|-|-|-|-|-|-|-
> CogVideoX|93.31|93.90|97.74|98.04|62.22|62.82|63.06|81.38
> VideoREPA (Ours)|93.46|93.95|97.76|98.05|61.39|62.72|63.29|81.38
>
> Model|Object Class|Multiple Objects|Human Action|Color|Spatial Relationship|Scene|Appearance Style|Temporal Style|Overall Consistency |Semantic Score (Overall)
> -|-|-|-|-|-|-|-|-|-|-
> CogVideoX|89.73|71.72|99.00|84.08|75.12|53.13|24.44|25.23|27.28|79.35
> VideoREPA (Ours)|89.60|71.40|98.20|85.37|76.03|53.05|24.49|25.24|27.28|79.45
>
> ### Questions
>
> > Q4.1. Is it fair to compare noisy activations from VDMs with the structure features of MAEs?
>
> Yes, we argue that this is a fair and informative comparison.
>
> While VDMs are trained for denoising, this process is itself a powerful form of self-supervised learning. Previous studies [3, 4] confirm that features from diffusion models are also structured and effective for downstream understanding tasks. **Therefore, evaluating these features for physics understanding is a valid approach.**
>
> Also, we build our evaluation protocol **following previous studies [5, 6]** which also evaluate the representation in diffusion models for understanding. Using a linear probing protocol for both, **we maximized the information extracted from the VDM's features evaluating different noise levels.** To our knowledge, this is a direct and appropriate way to probe the understanding capability of these specific representations.
>
> > Q4.2. Can you claim that large VDMs understand physics worse than smaller VFMs based on this unfair comparison (L47-50)?
>
> Given that our comparison is fair (as justified in our response to Q4.1), we can indeed claim that large VDMs understand physics worse than smaller VFMs: Our experiments showed a significantly weaker physics understanding for the large VDM; specifically, the VDM (2B) achieved 62% accuracy on the Physion (roll subset), while the much smaller VFM (86M) achieved 72%.
>
> This finding aligns with other studies [7, 8], which point out that VFMs learn powerful representations capable of performing physics understanding tasks and achieving strong results.
>
> > Q4.3. Doesn't the enhanced alignment is a trivial results by directly aligning the feature with MAE, rather than the results of truly enhanced physics understanding?
>
> No, the improvement is not a trivial result of feature alignment for two reasons.
>
> First, We show that the **improved physics understanding** (Fig. 1) **translates to enhanced physical plausibility in the generated videos** (Tab. 1). If the understanding improvement were merely an artifact of the evaluation setup, it would not lead to more physically realistic video, which is an independent task.
>
> Second, our qualitative results demonstrate a **comprehensive improvement across a wide range of physical phenomena**, including motion, causality, and complex interactions (details in answer to Reviewer vo9S, Q2). A trivial alignment would likely only improve superficial aspects, whereas our method achieves a broad and consistent enhancement of physical realism, suggesting a deeper, more fundamental transfer of physics knowledge.
>
> > Q5. Why does the spatial component of the TRD loss not exclude the case $i = j$, while the temporal component does exclude $d = e$?
>
> This design choice is intentional and reflects the distinct roles of the spatial and temporal components in our TRD loss:
>
> 1. The spatial component calculates the cosine similarity between all pairs of tokens $(i, j)$ **within a single frame** $d$. **When $i = j$:** The case represents the self-similarity of a token. It is 1, i.e., $y_{{spatial}}^{d,i,i} = \frac{{y} _ {d,i} \cdot {y} _ {d,i}}{||{y} _ {d,i}|| ||{y} _ {d,i}||} = 1$. Thus both the $y$ and $h$ terms equal to 1. The contribution to the TRD loss is always zero: $|h_{{spatial}}^{d,i,i} - y_{{spatial}}^{d,i,i}| = |1 - 1| = 0$.
>
>     Since these terms have no impact on the loss value and gradients, we kept them for notational and implementation simplicity.
>
> 2. The temporal component is designed specifically to capture the relationships between tokens **across different frames (i.e., $d\neq e$)**. Including the $d=e$ means calculating the similarity of tokens within the same frame, which is the **spatial component**, resulting double-counting.
>
>
> > Q6. The quantitative results show VideoREPA-5B outperforming HunyuanVideo, yet visually the latter appears better in quality. Why was HunyuanVideo not used as the baseline model?
>
> While HunyuanVideo may exhibit superior visual quality due to its larger scale, our research prioritizes **improving physical plausibility**, which is a distinct and critical challenge. Existing benchmarks like VideoPhy2 [9] show that CogVideoX outperforms HunyuanVideo in physical commonsense, as verified by human evaluation. Thus we chose CogVideoX as a stronger baseline.
>
> [1] Wan: Open and advanced large-scale video generative models[J]. arXiv:2503.20314, 2025.
>
> [2] Vbench: Comprehensive benchmark suite for video generative models[C]//CVPR 2024
>
> [3] Diffusion models as masked autoencoders[C]//CVPR 2023
>
> [4] Emergent correspondence from image diffusion[J]. NuerIPS 2023.
>
> [5] U-repa: Aligning diffusion u-nets to vits[J]. arXiv:2503.18414, 2025.
>
> [6] Representation alignment for generation: Training diffusion transformers is easier than you think[J]. arXiv:2410.06940, 2024.
>
> [7] IntPhys 2: Benchmarking Intuitive Physics Understanding In Complex Synthetic Environments[J]. arXiv:2506.09849, 2025.
>
> [8] Intuitive physics understanding emerges from self-supervised pretraining on natural videos[J]. arXiv:2502.11831, 2025.
>
> [9] Videophy-2: A challenging action-centric physical commonsense evaluation in video generation[J].arXiv:2503.06800, 2025.

---

> > ### Author Response · Authors · 2025-08-01
> >
> > Thank you for your detailed review. Regarding the most critical points you summarized, we provide a concise summary of our responses as follows:
> >
> > >**1. On the Justification for Our Method (Weakness 1):**
> >
> > We clarified that enhancing physics is the explicit goal of our work, not a byproduct. Our entire pipeline is motivated by the observed "physics understanding gap," and our novel TRD loss is specifically designed to transfer the relational knowledge inherent to physics. (See response to Q1)
> >
> > >**2. On the Validity of the Comparison in Figure 1:**
> >
> > Our comparison is fair because it is supported by recent work showing that features from denoising models are valid for understanding tasks, and it employs a consistent, standard protocol (linear probing) to assess the intrinsic knowledge in both models across different noise levels. We believe this establishes the validity of our findings. (See response to Q4.1)
> >
> > >**3. On a Broader Quantitative Comparison:**
> >
> > We conducted a comprehensive evaluation on the VBench benchmark. The results confirm that VideoREPA improves physical plausibility without compromising other key aspects of video quality. (See response to Q3)

---

> > > ### Comment · Reviewer_ryaw · 2025-08-02
> > >
> > > First, I appreciate your comprehensive feedback on my extensive reviews. Most of them make a lot of sense, except for the following points.
> > > I'll use your numbering policy for the response. Q3 is the only (and my last) question that you have to consider for raising the score.
> > >
> > > Q3. Your results suggest that VideoREPA couldn't enhance any score among all metircs in VBench. Can VBench not capture the physical plausibility of the generated videos? If so, I believe VBench2 can show the physical plausibility. (Mechanics, Thermotics, Material, and Multi-view consistency)
> > >
> > > Comments below are just minors:
> > > Q1. (minor, NO impact on the score) I fully understand your motivation and empirical results. As a result, TRD loss improves the physical plausibility of the generated videos in qualitative comparison & quantitative scores. For me, however, I couldn't find the explicit connection between TRD loss and physical plausibility. The TRD loss seems like just a soft alignment strategy between two video representation. I hope more clarification & justification would be added in your camera-ready version, if possible.
> > >
> > > Q5. (minor, NO impact on the score) If so, I think you'd better include those clarification or unify those expressions in the paper. It was a bit confusing for me.

---

> > > > ### Author Response · Authors · 2025-08-04
> > > >
> > > > Thank you for your valuable feedback and for giving us the opportunity to further clarify these points. We are encouraged to hear that addressing your final question could lead to a higher score.
> > > >
> > > > ### For the main questions:
> > > >
> > > > > Can VBench not capture the physical plausibility? And results on Mechanics, Thermotics, Material, and Multi-view consistency on VBench2.
> > > >
> > > > You have raised an important point. Yes, the VBench is **not** designed to measure physical plausibility. As noted by VBench2 [1], the focus of VBench is on aspects like video quality and condition consistency, while deeper commonsense and physics-based realism are beyond its scope.
> > > >
> > > > we conducted evaluations on the relevant physics-focused dimensions of the VBench2 [1]:
> > > >
> > > > | VBench2 Dimension | CogVideoX | **VideoREPA (Ours)** | Improvement |
> > > > | :--- | :---: | :---: | :---: |
> > > > | Material | 48.7 | **57.3** | **+8.6** |
> > > > | Mechanics | 49.1 | 49.1 | +0.0 |
> > > > | Thermotics | 54.0 | **54.7** | **+0.7** |
> > > > | Multi-view Consistency | 15.2 | **20.2** | **+5.0** |
> > > > | **Average** | 41.8 | **45.3** | **+3.5** |
> > > >
> > > > As shown, VideoREPA achieves a notable **+3.5 average improvement** across these physics dimensions. The gain is particularly significant in "Material" (which evaluates properties like gravity and buoyancy) and "Multi-view Consistency."
> > > >
> > > > We also observe that the improvement varies across different physical aspects—an interesting finding that points to the potentially deeper features in the proposed VideoREPA and suggests a promising direction for future, more targeted research.
> > > >
> > > > ### For the minor questions:
> > > >
> > > > > I hope more clarification & justification on the expicit connection between TRD loss and physics pluasibility would be added in your camera-ready version, if possible.
> > > >
> > > > Thank you for this feedback. We **will add more justification** about this in the next version of our paper. The core connection is that **intuitive physics is fundamentally relational** (how objects are configured spatially and evolve temporally), and our TRD loss is **specifically designed to distill these spatio-temporal relations**, making it a tailored strategy.
> > > >
> > > > > For the clarification on the formalization of TRD loss.
> > > >
> > > > We are glad our explanation resolved your initial confusion. We will ensure these clarifications are integrated into the final manuscript to improve readability. Thank you again for your constructive suggestions.
> > > >
> > > > **Reference**
> > > >
> > > > [1] Zheng D, Huang Z, Liu H, et al. Vbench-2.0: Advancing video generation benchmark suite for intrinsic faithfulness[J]. arXiv preprint arXiv:2503.21755, 2025.

---

> > > > > ### Comment · Reviewer_ryaw · 2025-08-04
> > > > >
> > > > > Thank you for your valuable comment. Most of my initial concerns were resolved during rebuttal & discussion period, so I decided to **increase my final score from 3 to 4**.
> > > > >
> > > > > The primary reasons for this decision are:
> > > > > 1. VideoREPA improves physics plausibility (VBench2) of the generated videos without compromising other features (VBench).
> > > > > 2. The comparison of noisy activations from VDM with MAE features is a fair, particularly when the diffusion timestep is zero (i.e. no noise is added).
> > > > >
> > > > > I appreciate your academic contributions and constructive discussion so far!

---

> ### Author Response · Authors · 2025-08-05
>
> **Thank you very much for your thoughtful reconsideration and for raising the score.** We are delighted that our responses and experiments successfully addressed your concerns. Your insightful feedback throughout this process has been invaluable in strengthening our work. We will extract the important suggestions and information and integrate them into the paper.
>
> We truly appreciate your constructive engagement. Please feel free to reach out if any further questions arise.

---

### Official Review · Reviewer_vo9S · 2025-07-03

**Clarity:** 3
**Significance:** 2
**Originality:** 3
**Rating:** 4
**Confidence:** 5

**Summary:**

This paper proposed a training strategy to advance the original video generation model with better physical consistency. It proposed an REPA learning strategy which aligns the feature from MM-DiT with the feature from a pretrained video model (e.g. VideoMAE).
VideoREPA is applied onto CogVideoX model and evaluated on two datasets: VideoPhy and VideoPhy2. On both semantic consistency and physical commonsense metrics, the proposed method performs better than original model.

**Questions:**

Please see the weakness part

**Ethical Concerns:**

["NO or VERY MINOR ethics concerns only"]

**Final Justification:**

The rebuttal has resolved my questions.

**Limitations:**

1. VideoREPA is a general technique which is compatible with all video generation foundation model. But the performance is only reported on CogVideoX model. It didn't show any generalization capability to other base models.

**Paper Formatting Concerns:**

1. The abbreviation VFM appears earlier than the full name video foundation model

**Quality:**

2

**Strengths And Weaknesses:**

Strengths
1. The proposed method shows better performance on both VideoPhy and VideoPhy2 datasets

Weakness
1. The performance gain from VideoREPA compared to original CogVideoX is very marginal, especially in the performance on SA (semantic adherence)

2. from the qualitative results, it seems the results from VideoREPA has more regular-shaped objects compared to original CogVideoX model, is this the reason why the physical related metrics could be improved? Because the objects in the generated videos have simpler shapes, leading to better physically interaction results

3. There is no physical performance evaluation on VideoMAE or VJEPA to validate why the features from MMDiT should be aligned to VideoMAE features.

4. The ablation study is not very comprehensive, e.g. it didn't explore different VFM but only uses VideoMAE. And it also didn't explore different alignment learning strategy but only tried REPA

---

> ### Author Rebuttal · Authors · 2025-07-31
>
> Thank you for taking the time to review our paper. We appreciate the chance to clarify our work and address your concerns, including any possible misunderstandings, in the responses below.
>
> ### Weakness
>
> > Q1. The performance gain from VideoREPA compared to original CogVideoX is very marginal, especially in the performance on SA (semantic adherence).
>
> Our VideoREPA improves PC significantly and maintain SA. This is reasonable **because the primary goal of VideoREPA is not to enhance video-text alignment (SA), but to improve Physical Commonsense (PC)**.
>
> As defined in the VideoPhy [1] and VideoPhy2 [2], PC and SA measure two distinct capabilities:
>
> *   **Physical Commonsense (PC):** Evaluates if the video's content (e.g., object motion, fluid dynamics, causality) adheres to real-world physical laws.
> *   **Semantic Adherence (SA):** Measures the alignment between the video content and the text prompt.
>
> The SA score is primarily monitored to ensure that our method does not achieve better physics at the cost of semantic fidelity by generating physically simpler, but less relevant, content.
>
> The results clearly demonstrate that VideoREPA achieves its primary goal. We observed a substantial **24.1% relative improvement in the PC**, while maintaining (or improving) the SA.
>
> Model|SA|PC
> -|-|-|
> CogVideoX-5B|70.0|32.3
> VideoREPA-5B (Ours)|72.1|**40.1 (+24.1%)**
>
> > Q2. from the qualitative results, it seems the results from VideoREPA has more regular-shaped objects compared to original CogVideoX model, is this the reason why the physical related metrics could be improved? Because the objects in the generated videos have simpler shapes, leading to better physically interaction results.
>
> Thank you for the question. While it is true that VideoREPA generates objects with more regular and stable shapes, which contributes to physical plausibility, this is only **one facet of the overall improvement** in PC. The PC evaluates a wide range of physical phenomena, and our qualitative results demonstrate that VideoREPA enhances many of these, far beyond just object shape.
>
> We highlight several examples below, **directly contrasting our results with the baseline** to demonstrate these multi-faceted improvements:
>
> *   **Causality and Interaction Logic:**
>     *   A glove correctly catches a baseball, unlike the baseline where the ball is incorrectly caught *behind* the glove (Fig. 1, row 1).
>     *   A glass shatters realistically upon hitting the floor, whereas the baseline shows unnatural motion (Fig. 1, row 2).
>     *   A crane maintains a direct physical connection while lifting bricks, unlike in the baseline (Fig. 3, row 2).
>     *   Detergent realistically mixes with water, causing the water level to rise, which is not depicted correctly in the baseline (Fig. 8, row 3).
>
> *   **Buoyancy:**
>     *   A bottle correctly floats on the sea's surface, while the baseline's bottle unnaturally lifts off the water (Fig. 6, row 3).
>
> *   **Light Reflection:**
>     *   A person using a hairdryer is accurately reflected in the mirror, correcting the distorted reflection seen in the baseline (Fig. 6, row 2).
>
> *   **Motion and Dynamics:**
>     *   Generating plausible rigid body motion, like a container rolling down a hill (Fig. 3, row 1) or a pencil rolling correctly (Fig. 3, row 2).
>     *   Modeling correct fluid dynamics, such as perfume spraying as a gas/fluid mixture (Fig. 7, row 3).
>     *   Adhering to principles like gravity (Fig. 9, row 2).
>
> *   **Shape Integrity:**
>     *   A credit card maintains its shape integrity when swiped, whereas the baseline's card partially vanishes (Fig. 6, row 1).
>
>
> Therefore, the enhanced physical realism stems not from merely generating correct shapes, but from a **comprehensive improvement in modeling plausible motion, causality, and interactions** that adhere to real-world physics. This is a direct result of our method's core mechanism: the **TRD loss**, which distills a deeper, relational understanding of both **spatial configurations (leading to better shapes and integrity) and temporal dynamics (leading to better motion and causality)** from the VFM.
>
> > Q3. There is no physical performance evaluation on VideoMAE or VJEPA to validate why the features from MMDiT should be aligned to VideoMAE features.
>
> The justification for aligning VDMs with VFMs like VideoMAEv2 is two-fold. First, as we demonstrate in our **Figure 1 evaluation on the Physion benchmark**, the VideoMAEv2 possess a **significantly stronger physics understanding ability** than the baseline VDM. This finding is also supported by recent studies [3, 4] on the emergent physics understanding capability in such models.
>
> Second, our work is based on the established principle that **enhanced understanding facilitates better generation** (Lines 151-166). Given the clear "physics understanding gap" we identified, we proposed VideoREPA to distill this superior understanding from the VFM into the VDM, thereby close understanding gap and enhancing the physical plausibility of the generated videos.
>
> > Q4.1. The ablation study is not very comprehensive, e.g. it didn't explore different VFM but only uses VideoMAE.
>
> We would like to clarify that an ablation study on different VFMs **was included in Appendix C (Table 4, Lines 550-553)**, where we compared VideoMAE, V-JEPA, OmniMAE, and VideoMAEv2.
>
> To further address your concern and to make our study more comprehensive, we have now included another powerful VFM, **V-JEPA 2 [5]**, and also explored various **combinations of VFMs**, as suggested by Reviewer CNDL. These new results not only demonstrate the flexibility of our VideoREPA framework but also provide deeper insights into how different VFMs contribute to performance (see answer to Reviewer CNDL Q2).
>
> |Models|SA|PC
> |-|-|-
> |- (baseline) |63.6|23.2
> |VideoMAE|59.8|26.2
> |V-JEPA|64.5 (2)|24.7
> |OminiMAE|61.6|24.7
> |OminiMAE+VideoMAE|62.5|25.6
> |OminiMAE+VideoMAE+VideoMAEv2|63.4|27.9 (2)
> |V-JEPA 2|65.1 (1)|27.3 (3)
> |VideoMAEv2|64.2 (3)|29.7 (1)
>
> *Numbers in parentheses indicate the rank (1st, 2nd, 3rd).
>
> *Note: With comparable SA scores, the PC score is the predominant metric for assessing the physics plausibility.*
>
> Lastly, we'd like to clarify a potential point of confusion: our primary VFM used throughout the paper is the powerful **VideoMAEv2, not VideoMAE**. VideoMAEv2's extensive pre-training on millions of videos makes it a significantly stronger foundation model for understanding physics than VideoMAE.
>
> > Q4.2. And it also didn't explore different alignment learning strategy but only tried REPA.
>
> Our contribution is not an direct application of REPA, **but the creation of VideoREPA, a novel framework with a new loss function (TRD loss)** designed specifically for physics knowledge transfer. We found REPA unsuitable as it is spatial-only and its "hard" feature alignment destabilizes pre-trained VDMs during finetuning (as shown in Fig. 4). Our "soft," relational TRD loss was developed precisely to overcome these critical failures.
>
> To the best of our knowledge, no other off-the-shelf alignment methods are designed for our task. Thus no available alternatives for ablation and the exploration of different alignment methods needs further consideration.
>
> However, we explored variants of our own strategy in Table 3. By ablating the spatial and temporal components, we evaluated different alignment approaches within our framework and confirmed that the full spatio-temporal design is the most effective.
>
> ### Limitations
>
> > Q5. The performance is only reported on CogVideoX model. It didn't show any generalization capability to other base models.
>
> Thank you for your suggestion. We mainly validated the effectiveness of VideoREPA on the CogVideoX model at two different scales (2B and 5B), demonstrating its strong performance and scalability in our paper. Following your advice, we have now applied VideoREPA to another powerful T2V model: Wan2.1 (1.3B) [6]. The results on the VideoPhy are shown below:
>
> Models|SA|PC
> -|-|-
> Wan2.1|67.1|21.5
> VideoREPA (ours)|66.9|**26.7**
>
> The results clearly show that VideoREPA significantly improves the PC score while maintaining SA. This provides empirical evidence for the generalization capability of our proposed method, demonstrating its effectiveness beyond the CogVideoX architecture.
>
> ### Paper formatting concerns
>
> > Q6. The abbreviation VFM appears earlier than the full name video foundation model.
>
> Thank you for the reminder. We initially abbreviated "Video Foundation Model" to VFM for clarity and to avoid repeating the lengthy phrase. However, given that VFM can also refer to "Vision Foundation Model" [7], we will change the abbreviation to ViFM in the revised manuscript to prevent any conceptual confusion. This change will be shown in next version of our paper.
>
> **Reference**
>
> [1] Bansal H, Lin Z, Xie T, et al. Videophy: Evaluating physical commonsense for video generation[J]. arXiv preprint arXiv:2406.03520, 2024.
>
> [2] Bansal H, Peng C, Bitton Y, et al. Videophy-2: A challenging action-centric physical commonsense evaluation in video generation[J]. arXiv preprint arXiv:2503.06800, 2025.
>
> [3] Bordes F, Garrido Q, Kao J T, et al. IntPhys 2: Benchmarking Intuitive Physics Understanding In Complex Synthetic Environments[J]. arXiv preprint arXiv:2506.09849, 2025.
>
> [4] Garrido Q, Ballas N, Assran M, et al. Intuitive physics understanding emerges from self-supervised pretraining on natural videos[J]. arXiv preprint arXiv:2502.11831, 2025.
>
> [5] Assran M, Bardes A, Fan D, et al. V-jepa 2: Self-supervised video models enable understanding, prediction and planning[J]. arXiv preprint arXiv:2506.09985, 2025.
>
> [6] Wan T, Wang A, Ai B, et al. Wan: Open and advanced large-scale video generative models[J]. arXiv preprint arXiv:2503.20314, 2025.
>
> [7] Wang W, Chen Z, Chen X, et al. Visionllm: Large language model is also an open-ended decoder for vision-centric tasks[J]. Advances in Neural Information Processing Systems, 2023, 36: 61501-61513.

---

> > ### Comment · Reviewer_vo9S · 2025-08-06
> >
> > Dear authors,
> >
> > Thank you for the detailed rebuttal response. I still have concerns about Q2 and Q3
> > On Q2, I am concerned the physical commonsense metric gain is obtained by sacrificing the details of the generated object, e.g. do we have any metric (aesthetic or image quality) to validate the quality of generated video is not degraded
> > On Q4, MLLM (multi-modal large language model) has better physical understanding capability than VideoMAE / VideoMAE v2, why don't we explore aligning the repsentation to models with better physical understanding?
> >
> > Look forward to your response

---

> ### Author Response · Authors · 2025-08-06
> **Looking forward to your further reply**
>
> Dear Reviewer vo9S,
>
> Thank you again for your time and for providing a detailed review of our work.
>
> We have submitted a comprehensive rebuttal where we aimed to address every one of your concerns, including:
> - **Clarifications on Key Points**: We clarified that improving PC is our primary goal, demonstrated comprehensive physics enhancements with qualitative examples, and highlighted the physical evaluation results on Video Foundation Models.
> - **Generalization to a New Model**: We applied VideoREPA to another T2V model (Wan2.1) to demonstrate its generalization capability.
> - **Expanded Ablation Studies**: We included results on an additional powerful Video Foundation Model and explored combinations of multiple models to make our ablations more comprehensive.
>
> We believe these clarifications and experiments provide a much stronger validation for our method.
>
> **As the discussion period is nearing its end**, we want to follow up and kindly ask if our responses have sufficiently addressed your questions. We are, of course, happy to answer any further questions you may have.
>
> Best regards,
>
> The Authors

---

> ### Author Response · Authors · 2025-08-07
>
> Dear Reviewer vo9S,
>
> Thank you for your continued engagement and for raising these important follow-up questions. We are happy to provide further clarifications.
>
> > Q2. Concerns about the physical commonsense metric gain is obtained by sacrificing the details of the generated object, e.g. do we have any metric (aesthetic or image quality) to validate the quality of generated video is not degraded.
>
> Thank you for this crucial question. To quantitatively verify that our improvements in physical commonsense do not come at the cost of visual quality, we conducted a **comprehensive evaluation on the VBench [1]**.
>
> The results, presented below, demonstrate that VideoREPA's performance on all quality and consistency metrics is **highly comparable to the baseline**. This provides strong evidence that **our VideoREPA enhances physical realism without degrading other essential aspects of video quality.**
>
> Model|Subject Consistency|Background Consistency|Temporal Flickering|Motion Smoothness |Dynamic Degree|Aesthetic Quality|Imaging Quality|Quality score (Overall)
> -|-|-|-|-|-|-|-|-
> CogVideoX|93.31|93.90|97.74|98.04|62.22|62.82|63.06|81.38
> VideoREPA (Ours)|93.46|93.95|97.76|98.05|61.39|62.72|63.29|81.38
>
> Model|Object Class|Multiple Objects|Human Action|Color|Spatial Relationship|Scene|Appearance Style|Temporal Style|Overall Consistency |Semantic Score (Overall)
> -|-|-|-|-|-|-|-|-|-|-
> CogVideoX|89.73|71.72|99.00|84.08|75.12|53.13|24.44|25.23|27.28|79.35
> VideoREPA (Ours)|89.60|71.40|98.20|85.37|76.03|53.05|24.49|25.24|27.28|79.45
>
> **(Note: The VBench is unable to measure physical realism [2].)**
>
> > Q4. MLLM (multi-modal large language model) has better physical understanding capability than VideoMAE / VideoMAE v2, why don't we explore aligning the repsentation to models with better physical understanding?
>
>
> This is an excellent and forward-looking question. We chose not to align with MLLMs for two primary, evidence-based reasons:
>
> **1. MLLM Representations are inherently Unsuitable for Relational Alignment:**
> The internal visual representations of MLLMs are fundamentally different from those of self-supervised (SSL) encoders. MLLMs [3, 4, 5] typically use a vision encoder (e.g., CLIP [6], SigLip-2 [7]) simply to project vision inputs into a shared space with text, with the subsequent layers trained for next text token prediction. **The visual tokens are training without explicit supervision in MLLMs** and there is currently no strong evidence that these intermediate visual tokens are structured in a way that is meaningful for alignment. **Aligning** with them directly **risks disturbing** the well-established feature space of the pre-trained video diffusion model.
>
>
> In contrast, SSL ViFMs (like VideoMAEv2, V-JEPA) are explicitly trained to produce semantically rich and structured latent representations that are proven to be effective for downstream tasks. Also the representation from Vision Foundation Models are proved to be beneficial when incoperating them into diffusion models [8, 9]. **Aligning with these well-structured and sematically rich features is a far more grounded approach**.
>
> **2. State-of-the-Art VFMs Outperform MLLMs on Physics Understanding Benchmarks:**
> Counterintuitively, current empirical evidence shows that specialized **VFMs actually demonstrate a stronger or comparable grasp of intuitive physics than general-purpose MLLMs.**
> *   On the **Physion** benchmark, **VideoMAEv2 achieves 72%** accuracy, significantly outperforming **GPT-4V's 52%** [10].
> *   Similarly, **V-JEPA achieves over 60%** on InfLevel-lab [11], while a powerful MLLM like **Qwen2-VL-72B performs at around the 50% (chance level)** [12].
>
> Therefore, based on current evidence, **ViFMs are not only more suitable for alignment but are also the superior source of physics knowledge.** While we agree that incorporating the reasoning of future, more capable MLLMs is a promising direction (**perhaps as a conditional guide by generating additional context-rich text tokens, rather than an alignment target**), our choice of VFMs is the most principled and effective one at present.
>
>
> Best regards,
>
> The Authors

---

> > ### Author Response · Authors · 2025-08-07
> >
> > **Reference**
> >
> > [1] Huang Z, He Y, Yu J, et al. Vbench: Comprehensive benchmark suite for video generative models[C]//Proceedings of the IEEE/CVF Conference on Computer Vision and Pattern Recognition. 2024: 21807-21818.
> >
> > [2] Zheng D, Huang Z, Liu H, et al. Vbench-2.0: Advancing video generation benchmark suite for intrinsic faithfulness[J]. arXiv preprint arXiv:2503.21755, 2025.
> >
> > [3] Liu H, Li C, Wu Q, et al. Visual instruction tuning[J]. Advances in neural information processing systems, 2023, 36: 34892-34916.
> >
> > [4] Wang P, Bai S, Tan S, et al. Qwen2-vl: Enhancing vision-language model's perception of the world at any resolution[J]. arXiv preprint arXiv:2409.12191, 2024.
> >
> > [5] Deng C, Zhu D, Li K, et al. Emerging properties in unified multimodal pretraining[J]. arXiv preprint arXiv:2505.14683, 2025.
> >
> > [6] Radford A, Kim J W, Hallacy C, et al. Learning transferable visual models from natural language supervision[C]//International conference on machine learning. PmLR, 2021: 8748-8763.
> >
> > [7] Tschannen M, Gritsenko A, Wang X, et al. Siglip 2: Multilingual vision-language encoders with improved semantic understanding, localization, and dense features[J]. arXiv preprint arXiv:2502.14786, 2025.
> >
> > [8] Yu S, Kwak S, Jang H, et al. Representation alignment for generation: Training diffusion transformers is easier than you think[J]. arXiv preprint arXiv:2410.06940, 2024.
> >
> > [9] Tian Y, Chen H, Zheng M, et al. U-repa: Aligning diffusion u-nets to vits[J]. arXiv preprint arXiv:2503.18414, 2025.
> >
> > [10] Venkatesh R, Chen H, Feigelis K, et al. Understanding physical dynamics with counterfactual world modeling[C]//European Conference on Computer Vision. Cham: Springer Nature Switzerland, 2024: 368-387.
> >
> > [11] Weihs L, Yuile A, Baillargeon R, et al. Benchmarking progress to infant-level physical reasoning in AI[J]. Transactions on Machine Learning Research, 2022.
> >
> > [12] Garrido Q, Ballas N, Assran M, et al. Intuitive physics understanding emerges from self-supervised pretraining on natural videos[J]. arXiv preprint arXiv:2502.11831, 2025.

---

> > ### Comment · Reviewer_vo9S · 2025-08-07
> >
> > Thanks for the response. The rebuttal has resolved my questions.

---

> ### Author Response · Authors · 2025-08-07
>
> Dear Reviewer vo9S,
>
> **Thank you for your thoughtful follow-up and for confirming that your concerns have been resolved**.
>
> We are very grateful for the discussion. Your rigorous questions have been invaluable in helping us clarify and strengthen our paper's contributions.
>
> Please feel free to let us know if any further questions arise.
>
> Best regards,
>
> The Authors

---

### Official Review · Reviewer_CNDL · 2025-07-03

**Clarity:** 3
**Significance:** 2
**Originality:** 3
**Rating:** 4
**Confidence:** 5

**Summary:**

This work proposes __VideoREPA__, a video diffusion model finetuning method which introduces physics knowledge into diffusion-based pretrained video generation models, where, instead of the token-level features, the relations between token features are aligned with those of a self-supervised video understanding foundation model using the RPEA loss. The proposed TRD loss aligns the pair-wise spatial and temporal token similarities. The experiments are conducted on two different datasets and the results demonstrates that the proposed VideoREPA is better at generating physical-commonsense video contents wrt the physical plausibility metrics.

**Questions:**

The questions have been summarized in the weakness part.

**Ethical Concerns:**

["NO or VERY MINOR ethics concerns only"]

**Final Justification:**

The authors response has addressed my concerns, where the extended experimental results are solid to verify the lacked analysis about the proposed method.

**Limitations:**

Yes.

**Paper Formatting Concerns:**

The formatting has no issue.

**Quality:**

3

**Strengths And Weaknesses:**

__Strengths__
1. the motivation is clear, where this work is motivated by the REPA in T2I methods and the information gap is analysed between video diffusion models and the self-supervised pretrained models.
2. the proposed solution takes the finetuning over pretrained VDM models into consideration, instead of directly use the original REPA loss, this work shifts it into spatial-temporal soft alignment. Note that the original REPA loss is introduced for accelerating training convergence.
3. the experimental results demonstrates empirical gains on two different datasets wrt the quantitative metrics.
4. the appendix is comprehensive, including implementation details and refinement strategy.


__Weakness__
1. This work highlights the improvement about the physical-commonsense, which is majorly validated with predefined metrics, however, considering the complexity of real world motions, these metrics may not accurately measure the plausibility of the generated motions. Detailed user study is required to demonstrated VideoREPA actually learns physical-commonsense.
2. The impacts of different VFMs seem significant (Table 4), is there any possibility to take use multiple VFMs to complement each other.
2. Are the shape of inputs of two models the same, considering the patch size of them may be different?

---

> ### Author Rebuttal · Authors · 2025-07-31
>
> We sincerely thank you for your thorough review and insightful comments. We are encouraged by your recognition of our paper's strengths, including its clear motivation, great novelty, strong performance and comprehensive writing. Your feedback is very helpful, and we address your questions below.
>
> > Q1. Detailed user study is required to demonstrated VideoREPA actually learns physical-commonsense.
>
> Thank you for your good suggestion. We conducted a user study to directly compare the performance of our **VideoREPA** against the **CogVideoX** baseline. For this study, we adopted the **Good-Same-Bad (GSB) pairwise comparison** method [1], which conducts pairwise comparisons to assess relative video quality. Participants were shown videos generated by both models for the 344 prompts from the VideoPhy dataset and were asked to evaluate them on two criteria:
> 1.  **Semantic Adherence (SA):** Which video better aligns with the text prompt?
> 2.  **Physical Commonsense (PC):** Which video appears more physically realistic (e.g., regarding causality, object motion, fluid dynamics)?
>
> The results are summarized in the preference table below:
>
> | User study | VideoREPA Wins | CogVideoX Wins | Tie |
> | - |-|-|-|
> | Semantic Adherence (SA) | 6.4% | 3.8% | 89.8% |
> | **Physical Commonsense (PC)** | **21.7%** | 8.0% | 70.3% |
>
> For physical realism, **our VideoREPA was preferred nearly 2.7 times more often than the baseline (21.7% vs. 8.0%)**. And VideoREPA and CogVideoX are comparable in semantic adherence, validating the improvement of PC is not at the cost of video-text alignment (SA).  This user study confirms that VideoREPA generates videos that are significantly more physically plausible.
>
> > Q2. The impacts of different VFMs seem significant (Table 4), is there any possibility to take use multiple VFMs to complement each other.
>
> Thank you for this insightful question. **The performance disparity observed in Table 4 likely stems from the scale of pre-training data and the resulting generalization capabilities of each VFM.** VideoMAEv2, which delivers the best performance (29.7 PC score), was pre-trained on millions of diverse, in-the-wild videos, enabling it to develop a robust, generalized understanding of physical dynamics. In contrast, models like VideoMAE and OmniMAE were trained on smaller datasets, leading to a comparatively weaker grasp of intuitive physics. We also **experimented with another powerful VFM, V-JEPA 2 [2]**, which also achieved a strong PC score of 27.3, reinforcing the idea that VFMs resulting from large-scale pre-training possibly have a better physical understanding.
>
> Following your suggestions, we **conducted new experiments to explore whether combining VFMs could yield complementary benefits.** We implemented this by applying a weighted average of the TRD losses from multiple VFMs. The results of these combination experiments are summarized below:
>
> |Models|SA|PC
> |-|-|-
> |- (baseline) |63.6|23.2
> |VideoMAE|59.8|26.2
> |V-JEPA|64.5 (2)|24.7
> |OminiMAE|61.6|24.7
> |OminiMAE+VideoMAE|62.5|25.6
> |OminiMAE+VideoMAE+VideoMAEv2|63.4|27.9 (2)
> |V-JEPA 2|65.1 (1)|27.3 (3)
> |VideoMAEv2|64.2 (3)|29.7 (1)
>
> *Numbers in parentheses indicate the rank (1st, 2nd, 3rd).
>
> Our findings are quite interesting:
>
> - Complementary Strengths: We found that combining models can indeed balance their individual strengths and weaknesses. For example, using only VideoMAE resulted in a lower SA score (59.8), while OmniMAE had a weaker PC score (24.7). Combining them (OmniMAE+VideoMAE) improved both SA (to 62.5) and PC (to 25.6) relative to their respective weak points.
>
> - Synergy with a Stronger Model: Adding the powerful VideoMAEv2 into the OminiMAE+VideoMAE further boosted performance, achieving a PC score of 27.9. While this is still lower than using VideoMAEv2 alone, it demonstrates that weaker VFMs can benefit from the combination of other stronger models. This opens a promising research direction: investigating whether advanced ensemble techniques or more sophisticated mixing strategies could yield a combined model that outperforms any individual constituent model.
>
> > Q3. Are the shape of inputs of two models the same, considering the patch size of them may be different?
>
> No, the input shapes for the two models are different. We address this misalignment through a multi-step process, as detailed in Lines 241-256 in the main paper and Lines 559-580 of our appendix.
>
> 1.  **Input Pre-processing for VFM:** The input for our base model, CogVideoX, is fixed at a high resolution (49 frames, 480x720). Feeding this directly into the VFM (e.g., VideoMAEv2) is computationally prohibitive. Therefore, we **uniformly downsample the video frames** to a lower resolution before passing them to the VFM. This strategy was chosen after exploring several trade-offs (Lines 568-572) and was found to best preserve the holistic temporal dynamics with manageable computational cost.
>
> 2.  **Handling Patch Size and Latent Misalignment:** The downsampled resolution for the VFM is chosen to be a multiple of its patch size to ensure compatibility. Even after this, a dimensional misalignment often remains between the VFM's feature outputs and the VDM's latent representations (due to different temporal compression ratios, etc.). To reconcile this, we **interpolate the VDM's latent features to match the VFM's feature dimensions** before calculating the TRD loss. This approach was empirically found to be the most effective (Table 5).
>
> **To be specific**, while CogVideoX processes the 49x480x720 video, the corresponding input for VideoMAEv2 is downsampled to 48x160x240. (Note we discard the first frame for the VFM alignment, as it primarily serves to maintain static semantic information in the VDM's 3D VAE and our focus is on distilling dynamic content (Lines 565-567)).
>
> [1] Gao Y, Guo H, Hoang T, et al. Seedance 1.0: Exploring the Boundaries of Video Generation Models[J]. arXiv preprint arXiv:2506.09113, 2025.
>
> [2] Assran M, Bardes A, Fan D, et al. V-jepa 2: Self-supervised video models enable understanding, prediction and planning[J]. arXiv preprint arXiv:2506.09985, 2025.

---

> ### Author Response · Authors · 2025-08-06
> **Looking forward to your further reply**
>
> Dear Reviewer CNDL,
>
> We sincerely thank you for your thoughtful and detailed feedback.
>
> We have carefully addressed your concerns in the rebuttal, including:
>
> - Additional user studies were conducted to validate that VideoREPA improved physics realism.
> - We explored the combination of multiple VFMs to complement each other and summarized insightful and interesting findings.
> - We illustrate the details for input shapes of two models.
>
> **As the discussion period is nearing its end**, we want to ensure that we have addressed all your concerns satisfactorily.
>
> Please feel free to reach out if you have any further questions — we’d be happy to discuss them with you.
>
> Best regards,
>
> Authors

---

### Note · Authors · 2025-08-12

We sincerely thank the reviewers for their detailed and valuable comments and the ACs for guiding the review process.

Over the discussion phases, we are pleased that our responses have resolved **all** concerns, leading to a consensus of positive outcomes:

*   **Reviewer CNDL**'s concerns were thoroughly addressed through an additional **user study** and **experiments combining multiple VFMs**, which revealed beneficial properties of our VideoREPA framework.

*   **Reviewer vo9S** confirmed that **"The rebuttal has resolved my questions,"** acknowledging our new experiments (on Wan2.1 and VBench) addressed all raised points.

*   **Reviewer ryaw** has **raised the score to positive**, confirming that our clarifications and VBench/VBench2 results resolved concerns about evaluation validity and video quality.

*   **Reviewer JMpo** validated our core motivation, stating our rebuttal was **"clear, concise and it addresses my 3 concerns."**

### Summary

The review process has affirmed the core merits of our work VideoREPA, a novel framework for distilling physics knowledge into text-to-video (T2V) models from video foundational models. The key contributions, validated by the reviewers, are:

**1. Clear Motivation & Novelty:** Our work is the first to identify and bridge the critical "physics understanding gap" between T2V models and VFMs. Our framework is novel in that the proposed TRD loss offers a "soft" relational alignment that overcomes the failures of prior methods like REPA for this task. (Acknowledged by Reviewers CNDL, ryaw, JMpo)

**2. Strong & Validated Performance:** VideoREPA achieves a substantial +24.1% improvement in Physical Commonsense (PC), with this quantitative gain supported by strong qualitative results and a new user study. (Acknowledged by Reviewers CNDL, vo9S, ryaw, JMpo)

**3. Comprehensive & Robust Evaluation:**
*   **No Quality Degradation:** Our VBench results confirm that gains in physics do not compromise video quality. (Acknowledged by Reviewers vo9S, ryaw)
*   **Proven Generalization:** We demonstrated our method's effectiveness on another powerful model (Wan2.1). (Acknowledged by Reviewers vo9S, ryaw)
*   **Thorough Implementation:** Our comprehensive implementation details stated in the paper were also noted as a strength. (Acknowledged by Reviewers CNDL, ryaw)

Overall, the reviewers' feedback demonstrates a strong positive consensus on the paper's contributions.

---

### Decision · Program_Chairs · 2025-09-17

**Decision:**

Accept (poster)

**Comment:**

This paper proposes VideoREPA, a work designed to bridge the “physics understanding gap” in text-to-video diffusion models by distilling relational knowledge from video foundation models (VFMs). The central contribution is the Token Relation Distillation (TRD) loss, which aligns spatial and temporal token relations rather than enforcing hard feature-level alignment. This approach directly targets the challenge of physical plausibility in generated videos and is supported by clear motivation, validated improvements, and demonstrated method design. The work is well-aligned with current trends, given increasing interest in improving controllability and realism in video generation.

Across the review process, the authors provided several clarifications and experiments that addressed initial concerns. They conducted a user study demonstrating clear improvements in physical realism, explored combinations of multiple VFMs to show complementary strengths, and validated generalization by applying VideoREPA to Wan2.1 in addition to CogVideoX. Further, evaluations on VBench and VBench2 confirmed that improvements in physics plausibility were achieved without compromising video quality or semantic adherence. These additions strengthened the robustness, and all reviewers acknowledged that their concerns had been resolved.

In sum, the paper makes a well-motivated and technically contribution by introducing a novel relational distillation strategy for enhancing physical common sense in video generation. While some limitations remain, the overall work is convincing and impactful. The consensus after discussion is positive, and I recommend acceptance.